# Controlling light in complex media beyond the acoustic diffraction-limit using the acousto-optic transmission matrix

Ori Katz[1,2,3], François Ramaz[2], Sylvain Gigan [3] & Mathias Fink [2]

Studying the internal structure of complex samples with light is an important task but a difficult challenge due to light scattering. While the complex optical distortions induced by scattering can be effectively undone if the medium's scattering-matrix is known, this matrix generally cannot be retrieved without the presence of an invasive detector or guide-star at the target points of interest. To overcome this limitation, the current state-of-the-art approaches utilize focused ultrasound for generating acousto-optic guide-stars, in a variety of different techniques. Here, we introduce the acousto-optic transmission matrix (AOTM), which is an ultrasonically-encoded, spatially-resolved, optical scattering-matrix. The AOTM provides both a generalized framework to describe any acousto-optic based technique, and a tool for light control and focusing beyond the acoustic diffraction-limit inside complex samples. We experimentally demonstrate complex light control using the AOTM singular vectors, and utilize the AOTM framework to analyze the resolution limitation of acousto-optic guided focusing approaches.

[1] Department of Applied Physics, The Hebrew University of Jerusalem, 9190401 Jerusalem, Israel. [2] Institut Langevin, ESPCI Paris, CNRS UMR7587, PSL Research University, 1 rue Jussieu, 75005 Paris, France. [3] Laboratoire Kastler Brossel, Ecole Normale Supérieure, Collège de France, CNRS UMR 8552, Sorbonne université, 24 rue Lhomond, 75005 Paris, France. Correspondence and requests for materials should be addressed to O.K. (email: orik@mail.huji.ac.il) or to M.F. (email: mathias.fink@espci.fr)

Conventional optical focusing and imaging techniques fail in strongly scattering media because of the multiple-scattering events that any incident optical beam undergoes in its propagation inside them. However, the complex wavefront distortions, even deep inside diffusive samples, can be effectively reversed by high-resolution shaping of the input optical wavefront[1], in a fashion analogous to time-reversal experiments in ultrasound[2]. The ability to digitally control optical interference in multiply-scattering media using spatial light modulators (SLMs) has recently given rise to new focusing and imaging techniques[3,4]. Following the pioneering work of Vellekoop and Mosk[5], wavefront optimization was used for correcting spatial[6–9], temporal[10,11], spectral[12], and polarization[13] distortions, and to optimize transmission[14] through multiply-scattering media. A generalized theoretical framework underlying all experiments involving light propagation in complex media is the reflection/transmission matrix (TM) formalism[6,15–17]. The TM essentially contains the medium's response at every output spatial mode to excitation by any input spatial mode, i.e., the medium's set of Green functions. Experimental access to the optical TM was first made possible by measuring the response from each pixel of an SLM placed at a chosen input plane to each pixel of a camera placed at a desired output plane[6]. However, while such experimentally measured TM allowed focusing and imaging, this could only be done at the camera output plane, located outside the scattering sample. Thus, severely limiting its practical use for non-invasive imaging inside complex samples, where the goal is to perform imaging without direct access to the target plane. To control scattered light inside scattering samples a "guide star" providing feedback on the optical intensity at the target point is required[4]. While the singular value decomposition (SVD) of an all-optical TM was used to focus light on isolated reflective targets[18–20], such all-optical approaches are not capable of focusing light unless an isolated, strongly reflective target is present[16,18]. Moreover, even in the presence of such a reflective target, the obtained focus is of the target dimensions, and not the desired optical diffraction limit[20]. The requirement for an isolated reflective target can be overcome using advanced mechanisms for guide-stars, based on non-linear particles[21–23], photoacoustic feedback[24–26], and acousto-optic tagging[27–33]. The last two techniques combine light and sound to benefit from the near scattering-free propagation of ultrasound in optically scattering samples. In photo-acoustics, the interaction of light with absorbing parts of the sample generates ultrasound waves, which allows ultrasonic imaging and optical focusing on optical absorbers. In acousto-optics, the requirement for optical absorption is removed by locally modulating the diffused light inside the sample using a focused ultrasound beam[34]. Detection of the acoustically modulated (frequency shifted) light enables measuring only the light that has traveled through the ultrasonic focal spot, with a spatial resolution given by the ultrasound focus dimensions. Light can also be focused back into the ultrasound focus via optical phase-conjugation of the tagged light in "time-reversed ultrasonically encoded" (TRUE) optical focusing[27,28,30], or via iterative optimization[33]. Acousto-optic tagging holds many advantages, being a non-invasive, three-dimensional (3D)-positionable, label-free, non-ionizing guide-star. Its main drawback, however, is that the spatial dimensions of the ultrasound focus are given by the ultrasonic wavelength, yielding a resolution that is orders of magnitude inferior than the optical diffraction limit[4].

To date, two approaches to overcome the acoustic diffraction limit in acousto-optics have been put forward: iterative TRUE (iTRUE)[31,32], and time reversal of variance-encoded (TROVE) optical focusing[29]. In TROVE the intensity fluctuations of ultrasonically tagged light for different random inputs are analyzed, and an optical wavefront that focuses to the location with increased fluctuations is computed, allowing, in principle, optical speckle size focusing[29]. In iTRUE, multiple iterations of phase-conjugation are used to improve the focusing resolution[31,32]. At each iteration the optical beam is refocused to the ultrasonically tagged region, modulated again by the focused ultrasound, and thus shrinks progressively. $N$ iterations of iTRUE are required for a $\sqrt{N}$ resolution increase beyond the acoustic diffraction limit. Both iTRUE and TROVE rely on a digital optical phase-conjugation (DOPC) system[35], a rather complex apparatus, which conjugates a high-resolution SLM and a camera. DOPC systems requires a very precise pixel-to-pixel alignment that can be experimentally challenging[35]. Nonetheless, DOPC is until now the only tool that allows to overcome the acoustic diffraction limit in acousto-optic guided optical focusing.

Here, we introduce a novel generalized concept for light control using acousto-optic guidance: the acousto-optic transmission matrix (AOTM). Our concept is based on the understanding that any experiment that utilizes linear ultrasound tagging can be described by a single linear operator, and that this operator can be described by a single matrix. We show that a single AOTM provides a general, concise, and full description of light propagation in any acousto-optic experiment. We experimentally demonstrate how the AOTM can be measured using a single or multiple ultrasonic focused beams, and how its SVD allows for sub-acoustic optical focusing and light control inside a complex medium, without a DOPC system. We use the AOTM to show how all present acousto-optic techniques, namely TRUE, iTRUE, and TROVE, can be described under the same framework, and how the AOTM enables performing TRUE, iTRUE, and TROVE experiments without a DOPC system, i.e., using any arbitrary positioning of an SLM and a camera. We also utilize the AOTM framework to analyze the resolution limitation of SVD-based acousto-optic guided focusing approaches.

## Results

**Principle of the AOTM with a single ultrasonic beam.** Consider a general acousto-optical experiment, such as the one schematically depicted in Fig. 1a, where diffused, quasi-monochromatic light, is ultrasonically tagged by a focused ultrasound beam, and subsequently measured by a camera placed outside the medium. Assuming linear light propagation and linear acousto-optic interaction, the relationship between any input optical field $E^{in}(f_o)$ at the optical frequency $f_o$, and the ultrasonically tagged output field at the camera plane, $E^{out}(f_o + f_{US})$, which is frequency shifted to a frequency $f_o + f_{US}$ by the ultrasound beam at frequency $f_{US}$, is given by a linear operator $T_u$:

$$E^{out}(f_o + f_{US}) = T_u E^{in}(f_o) \qquad (1)$$

where $u$ is an index describing the ultrasound focus position. We define the matrix describing this operator in the spatial domain as the AOTM. Specifically, each element of the AOTM, $t_{u,mn}$, gives the complex amplitude of the acoustically tagged optical field at the output spatial position $r_m$, $E_m^{out}$, as result of an input field at position $r_n$, $E_n^{in}$, and an ultrasound modulating focus positioned at $r_u$ (an acousto-optic "Green function"). Importantly, $r_m$ is located at the camera plane, outside the scattering medium, and $r_u$ is located inside the medium. As a result of linearity, the output acoustically tagged field measured at $r_m$ for a general input field is:

$$E_m^{out}(f_o + f_{US}) = \sum_n t_{u,mn} E_n^{in}(f_o) \qquad (2)$$

where the summation is done over all spatial input modes, $n$. Fig. 1a depicts the basic setup required for acquiring the AOTM and subsequently exploiting it for light control. The setup is based

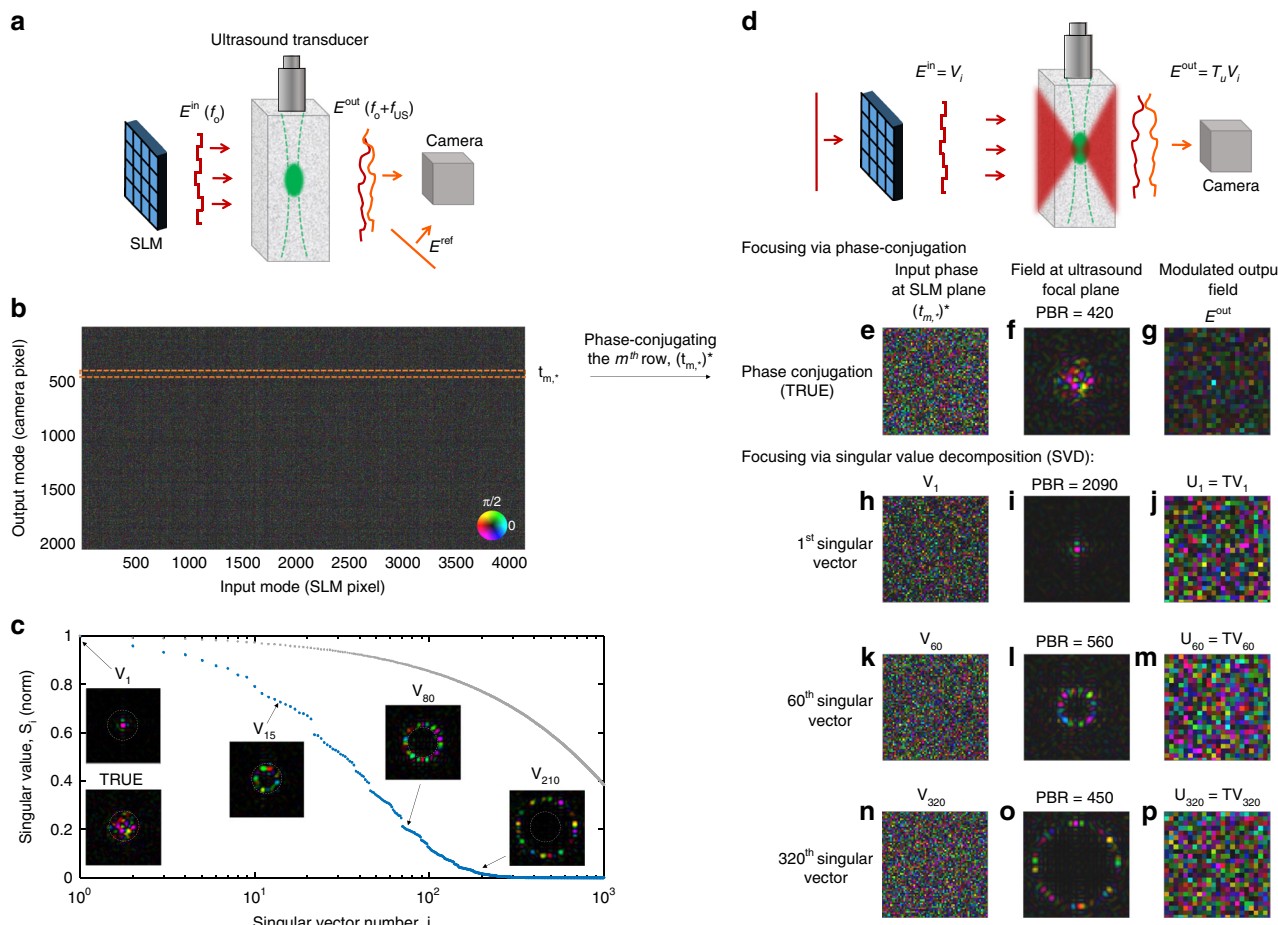

**Fig. 1** Acousto-optic transmission-matrix simulated acquisition and use. **a** Considered setup for acquiring the acousto-optic transmission matrix (AOTM): The AOTM is acquired by injecting a basis of optical modes $E_n^{in}(f_o)$ at the laser frequency $f_o$ into a multiply-scattering sample using a spatial light modulator (SLM), and recording the ultrasound modulated light outside the sample, $E_m^{out}(f_o + f_{US})$, with a camera, using a pulsed reference beam at the ultrasound shifted frequency $E^{ref}(f_o + f_{US})$. The ultrasound modulation is provide by a pulsed ultrasound beam at a frequency $f_{US}$ that is focused inside the sample. The scattered light is assumed to be a multiply-scattered diffused field having no ballistic components or speckle correlations. **b** An AOTM as obtained from a simulated acquisition, **c** first 1,000 singular values of the AOTM (blue), which are dictated by the ultrasound focus shape, vs. the first 1,000 singular values of an all-optical TM measured without ultrasound tagging (gray), which follows the general Marčenko-Pastur distribution, insets: optical fields at the ultrasound focal plane when the corresponding singular vectors of the AOTM are injected into the medium (as would be observed by removing half of the scattering medium between the focus and the camera, see Fig. 2); **d–p** using the AOTM for focusing and light control: **e–g** phase-conjugating the $m^{th}$ row of the AOTM is equivalent to time reversal of ultrasound encoded light (TRUE) focusing, concentrating the optical intensity to the ultrasound focus; **h–j** injecting the AOTM first singular vector focuses light at the center of the ultrasound focus, with a resolution beyond the acoustic diffraction limit, equivalent to infinite iterations of iterative TRUE; **k–p** injecting singular vectors with lower singular values result in intensity enhanced rings with increasing diameter around the center of the ultrasound focus. Low singular values concentrate energy outside the ultrasound focus (**n–p**). (Numerical simulations in the diffusive multiple-scattering light propagation regime)

on the well-established approach for measuring the optical transmission matrix (TM)[6], with the addition of an ultrasound transducer that generates an ultrasound pulsed focus inside the medium, and a pulsed optical reference arm for off-axis holography. The setup is composed of an illuminating laser beam that passes through a computer-controlled SLM to provide injection of controlled optical modes into the medium. The output acoustically tagged scattered fields at a frequency $f_o + f_{US}$ are measured outside the medium via off-axis, phase-shifting holography[36], using the pulsed reference beam that is synchronized with the ultrasound pulses (see Methods).

The AOTM is acquired by sequentially injecting each of the $n = 1..N_{SLM}$ input modes, $E_n^{in}(f_o)$, having the laser frequency $f_o$ into the medium (where $N_{SLM}$ is the number of SLM pixels used), and measuring the frequency-shifted output field $E_m^{out}(f_o + f_{US})$ at each of the $m = 1..M$ camera pixels simultaneously. Following Equation (2), in each of the $n = 1..N_{SLM}$ measurements steps, the

$n^{th}$ column of $T_u$ is acquired, giving a matrix of dimensions $M \times N_{SLM}$ (Fig. 1b). The measured AOTM thus describes the propagation of light from the input plane of the SLM, through the small ultrasound focus inside the medium, to the camera plane. While an all-optical TM that is measured without the ultrasound focus[6], reflects the combined interference of all optical paths inside the scattering medium, using the AOTM, the spatially localized ultrasound focus allows spatially resolved probing and focusing in a specific target volume inside the medium, as we show below.

Once the AOTM is acquired (as is simulated in Fig. 1 using a random independent and identically distributed (i.i.d.) scattering matrix model for a multiply-scattering medium, see Methods), it can be used for optical focusing inside the medium by several different approaches (Fig. 1d–p). The most straightforward but least powerful approach is via direct phase-conjugation: displaying the phase conjugate of the $m^{th}$ row of the AOTM on the SLM

(dashed orange box in Fig. 1b, Fig. 1e), leads to light focusing at the $m^{th}$ camera pixel (Fig. 1g), and as result of the ultrasound tagging, also to light concentrating inside the acoustic focus (Fig. 1f). The dimensions of the intensity-enhanced volume inside the sample are the dimensions of the ultrasound focus, as would be obtained in TRUE focusing experiments[27,28,30]. This is not a coincidence but rather a deeper result obtained from optical reciprocity: phase-conjugating an AOTM row mathematically describes a TRUE experiment with a virtual initial illumination at the $m^{th}$ camera pixel (see Supplementary Note 2). The peak to background ratio (PBR) of the formed focus at the ultrasound focus plane inside the scattering medium is also the same as theoretically expected from TRUE focusing, i.e., $PBR \approx N_{SLM}/N_s$, where $N_s$ is the number of speckle grains contained in the ultrasound focus. The PBRs obtained in the simulations of Fig. 1 with a phase-only SLM having $N_{SLM} = 4096$ pixels are in good accordance with these expected values.

A considerably more powerful optical focusing and control approach that allows surpassing the acoustic diffraction limit, is via SVD of the AOTM. SVD is a powerful tool in matrix analysis, which was recently used to identify transmission eigenchannels[37–40], and for selective focusing[18,25]. The main interest in the SVD of the AOTM is that the first singular vector of the AOTM gives the optimal wavefront for the tightest optical focusing, as would be obtained after infinite iterations of iTRUE[31,32]. This important result is proved in detail in Supplementary Note 3. A short intuitive explanation is that since the AOTM, $T_u$, describes a single pass through the medium and ultrasound focus, the matrix $T_u^H T_u$ (where $H$ is the Hermitian conjugate) describes two iterations of phase-conjugation of iTRUE going back and forth through the ultrasound focus. Performing $2k$ additional iterations of iTRUE is then mathematically described as taking the matrix $T_u^H T_u$ to $k$-th power. Thus, iTRUE iterations are in fact power iterations of the matrix $T_u^H T_u$. Since performing an infinite number of power iterations is the mathematical approach for finding the largest eigenvalued eigenvector of $T_u^H T_u$. This eigenvector is, by definition, the first singular vector of the AOTM, $T_u$, providing the sharpest optical focus. The fact that the SVD of the AOTM provides sharp focusing is due to the gaussian-like shape of the ultrasound tagging focus. This is in contrast to the SVD of the all-optical TM (Fig. 1c, gray dots), which yields transmission eigenchannels whose injection into a scattering medium does not lead to spatial focusing[20,38].

Numerical results for injecting the first singular vector of the AOTM show the formation of a tightly focused spot at the center of the acoustic focus (Fig. 1h–j). The focus size reaches the optical diffraction limit (i.e., a single optical speckle grain) if a sufficiently large number of input modes, $N_{SLM}$, are measured, where the required $N_{SLM}$ is larger for larger ultrasound focus size, or smaller speckle grain size (see Supplementary Notes 5–6). Importantly, not only the first singular vector of the AOTM is of interest: singular vectors with decreasing singular values will form concentric rings with increasing diameter around the center of the ultrasound focus (Fig. 1c insets, Fig. 1k–p). Using low singular values leads to concentration of energy outside the acoustic focus, in a fashion resembling open channels in systems containing localized absorption[40]. This resemblance to open channels may be understood from the fact that the singular values of the AOTM represent the energy transmission through the acoustic focus. The distribution of singular values of the AOTM (Fig. 1c—blue dots) is dictated by the shape and size of the acoustic focus, and is very different than the general Marčenko–Pastur distribution of the all-optical TM (Fig. 1c—gray dots). The number of significant singular values of the AOTM is approximately the number of optical modes (speckles) contained inside the acoustic focus, since it is the number of optical modes that decompose the effective

virtual "aperture" that is formed by the ultrasound focus. For the Gaussian-shaped ultrasound focus considered in Fig. 1 the singular values decrease gradually. Different shaped ultrasound foci would result in different distributions (see a numerical example in Supplementary Note 4).

**AOTM using a single ultrasonic beam**. To experimentally demonstrate our approach we used the setup schematically described in Fig. 2a, (see Methods and Supplementary Note 1). As was used in several recent works[29,41,42], in our proof-of-concept experiments the sample was made of two optical diffusers separated by a distance to allow access to the focusing plane. A pulsed focused ultrasound transducer with a central frequency of 15 MHz placed perpendicular to the light propagation direction was used for acousto-optic tagging. Similar to recent works[29,42], a relatively large speckle grain size ($\sim 30\,\mu m$) compared to the optical wavelength was chosen as a convenient proof of principle for single-speckle grain focusing using the few thousands of input modes, $N_{SLM}$, that can be experimentally measured with our setup within the samples decorrelation time. Achieving single-speckle grain focusing with smaller sized speckles, requires larger $N_{SLM}$ (see Supplementary Notes 5–6). However, a too small $N_{SLM}$ would still provide a focus size that is smaller than the ultrasound diffraction limit and a higher PBR than TRUE (see Supplementary Figures 6–9). Fig. 2 presents the experimental results of optical control using the SVD of a measured AOTM. Supplementary Figure 10 displays the AOTM measured in this experiment and the distribution of its singular values. As expected, injecting the first singular vectors generates a sharp optical focus with dimensions smaller than the acoustic focus (Fig. 2c, d). Injecting singular vectors with lower singular values results in the formation of concentric rings of increased optical intensity around the center of the ultrasound focus (Fig. 2e, f). Using low singular values leads to concentration of energy outside the acoustic focus (Fig. 2g–i). The full-width at half-max (FWHM) transverse horizontal dimensions of the ultrasound focus (Fig. 2b) and the sharpest focus formed via SVD of the AOTM (Fig. 2c) are $170\,\mu m \pm 10\,\mu m$ and $35\,\mu m \pm 5\,\mu m$, respectively. The vertical dimensions of these foci (along the ultrasound axial dimension) are $175\,\mu m \pm 20\,\mu m$ and $35\,\mu m \pm 5\,\mu m$, respectively. Thus, the SVD of the AOTM provides here a resolution increase beyond the acoustic diffraction limit of approximately 4.8. We note that the AOTM approach is general and is not limited to thin scattering layers. This is proved numerically in Fig. 1, where light propagation in the multiple-scattering diffusive regime was simulated, by considering random i.i.d. matrices for modeling the light propagation in a random multiply-scattering sample (see Methods).

**AOTM using two ultrasound beams**. The first singular vector of a single AOTM (Figs. 1–2) allows focusing only at a single point located at the center of the ultrasonic focus. Given the large number of measurements required for such focusing, this forms a limitation for the use of this approach for applications such as imaging. However, as was recently shown by Judkewitz et al. in their TROVE work[29], a joint analysis of several matrices of acousto-optically modulated output modes, measured for different ultrasound foci positions, can allow focus scanning. Here, we apply the same mathematical approach to jointly decompose multiple AOTMs measured for two different ultrasound focus position. Specifically, we exploit the joint analysis of two AOTMs to perform a scan of a tight optical focus over multiple positions. For this purpose, we consider the separate measurement of two AOTMs, $T_1$ and $T_2$, using two different ultrasound focused beam that are spatially shifted such that the two ultrasound focal spots,

$P_1(x,y)$ and $P_2(x,y)$, partially overlap (Fig. 3a–d). Such ultrasound focal spots are easily obtained in the same experimental setup presented in Fig. 2 by changing the time-delay, $\Delta\tau = z/v_{US}$, between the ultrasound pulse and the optical pulses (Fig. 3a), where $z$ denotes the axial distance of the acoustic focal spot from the ultrasound transducer, and $v_{US}$ is the speed of sound in the medium. Injecting the first singular vector of $T_1$ or $T_2$ would only form a tight optical focus at the center of each of the two ultrasound foci. However, the joint information in the two matrices can be exploited to scan even a sharper focus along the axis connecting the centers of the two acoustic foci (Fig 3i–k).

Noting that the SVD of a single AOTM, $T$, is obtained by diagonalizing the operator $T^H T$, referred in ultrasound experiments as the time-reversal operator (TRO)[43–45], and following the multi-focus analysis of TROVE[29], we used the first eigenvectors or singular vectors of the following generalized TRO, $A_\alpha$, to form a focus at different controlled positions:

$$A_\alpha = \left[(T_1 - \alpha T_2)^H (T_1 - \alpha T_2)\right]^{-1}\left[(T_1 + \alpha T_2)^H (T_1 + \alpha T_2)\right] = (T_{1-\alpha 2})^{-1} T_{1+\alpha 2} \tag{3}$$

where $\alpha$ is a positive weighting parameter controlling the focus position. $\alpha = 1$ yields focusing at the middle of the line connecting the two ultrasound foci centers (Fig. 3i). The matrix $A_\alpha$ is the multiplication between the TRO of the weighted sum of $T_1$ and $T_2$: $T_{1+\alpha 2} = (T_1 + \alpha T_2)$ (see Fig. 3d), by the inverse of their weighted difference: $T_{1-\alpha 2} = (T_1 - \alpha T_2)$. Scanning the tight focus is made possible because, as result of linearity, the difference matrix: $T_{1-2} = (T_1 - T_2)$ describes an AOTM of a virtual

ultrasound focus that is the subtraction of the first ultrasound focus pressure field from the other (Fig 3e). This difference acoustic pressure field is equal to zero at a specific distance between the two ultrasound foci centers (Fig. 3e). Dividing the sum of the two ultrasound foci (Fig. 3d) by their difference (Fig. 3e) results in a sharp peak at this distance (Fig. 3f). The position of the sharp peak is controlled by the parameter $\alpha$. In practice, to take into account measurement noise, the matrix inversion in the calculation of $A_\alpha$ is performed via a pseudo-inverse ($^+$) with a proper regularization parameter[46].

Figure 3i–k present numerical results obtained with this approach in a multiply-scattering medium, along with a comparison to TRUE focusing (Fig. 3g, h). Fig. 4 presents experimental results of a proof-of-principle experiment with two scattering layers. It can be noticed that focusing using the joint information of the two matrices (Fig. 4e–h) yields a sharper focus than that obtained via SVD of each matrix separately (Fig. 4d). This is made possible since the virtual ultrasound focus obtained by dividing by pressure difference $T_{1-\alpha 2}$ (Fig. 3f) is smaller than each of the single ultrasound foci (Fig. 3b, c). This result can be extended to allow scanning in two or three dimensions using more ultrasound beams, as was demonstrated by Judkewitz et al. using four ultrasound foci[29].

Interestingly, the analogy between the results obtained by SVD of the AOTM and those obtained with TROVE is not coincidental. The underlying reason is that the variance maximization approach of TROVE is, in fact, based on performing an SVD to a measured matrix of acoustically modulated output modes that is very similar to the AOTM: in

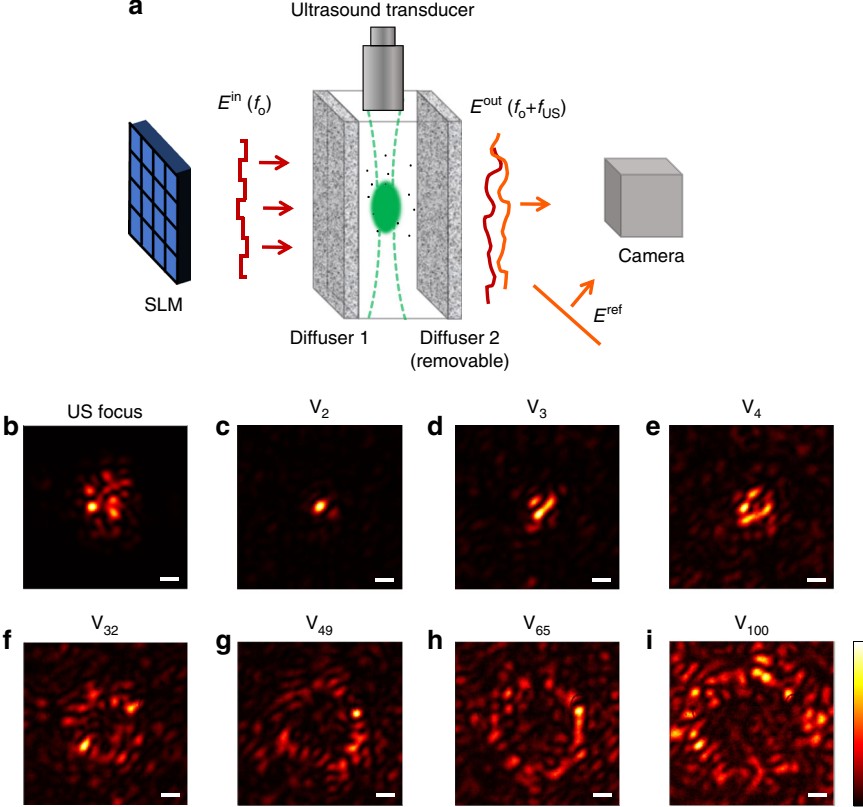

**Fig. 2** Focusing using singular value decomposition of an acousto-optic transmission matrix. **a** Experimental setup. **b–i** measured optical intensity distributions at the ultrasound focal plane: **b** optical intensity of the ultrasonically tagged optical field for a plane wave input, displaying the dimensions of the ultrasound focus. **c–i** optical intensity distributions measured when injecting the 2nd, 3rd, 4th, 32nd, 49th, 65th, and 100th singular vectors of the acousto-optic transmission matrix (AOTM). While the first singular vectors focus light at a sharp focus smaller than the ultrasound focus, with peak to background ratio (PBR) of 290, 175 and 94 for **c**, **d**, and **e** respectively, singular vectors with lower singular values result in concentric rings around the ultrasound focus center with increasing diameter. scale-bars, 67 μm

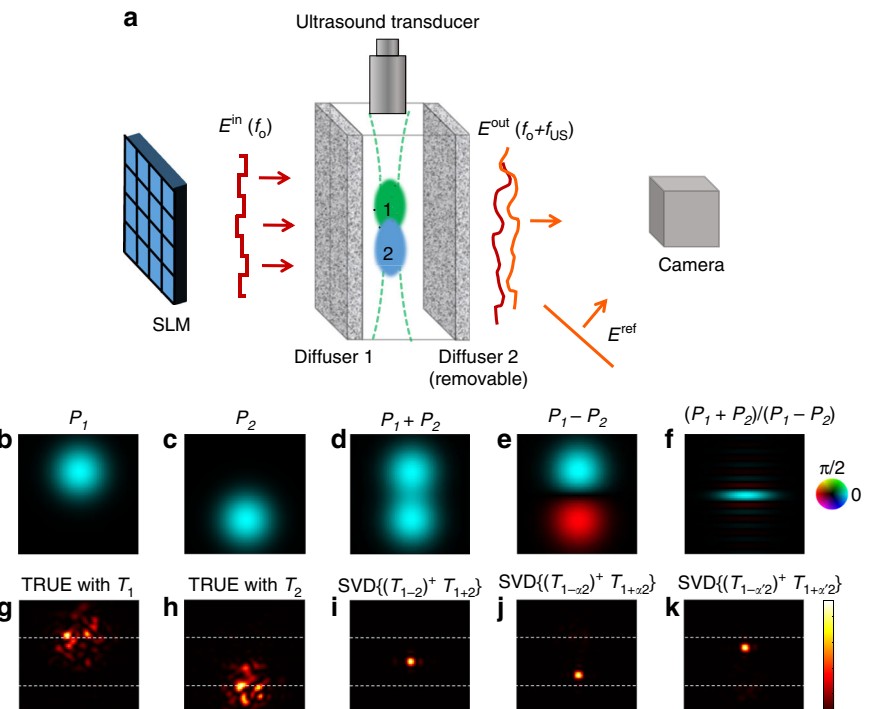

**Fig. 3** Simulations of focus scanning using two acousto-optic transmission matrices. **a** Measurement setup: two partly overlapping ultrasound foci (marked by "1" and "2" in green and blue, respectively) are used to measure two acousto-optic transmission matrices (AOTMs), $T_1$ and $T_2$; **b, c** ultrasound pressure amplitude distributions for the first ($P_1$) and second ($P_2$) focus; **d, e** sum and differences of the ultrasound pressure fields; **f** sum of the pressure fields divided by the difference between the pressure fields, displaying a sharp peak between the two ultrasound foci centers; **g–k** optical intensity distributions at the ultrasound focal plane: for time reversal of ultrasound encoded light (TRUE) focusing using $P_1$ or $P_2$ (**g, h**), and focusing via singular value decomposition (SVD) of the matrix $A^\alpha$ (Equation 3) formed by the weighted sum of $T_1 + \alpha T_2$ divided by their weighted difference $T_1$-$\alpha T_2$, for different values of $\alpha$: $\alpha = 1$ (**i**), $\alpha = 0.05$ (**j**), and $\alpha = 20$ (**k**). (Numerical simulations in the diffusive multiple-scattering light propagation regime)

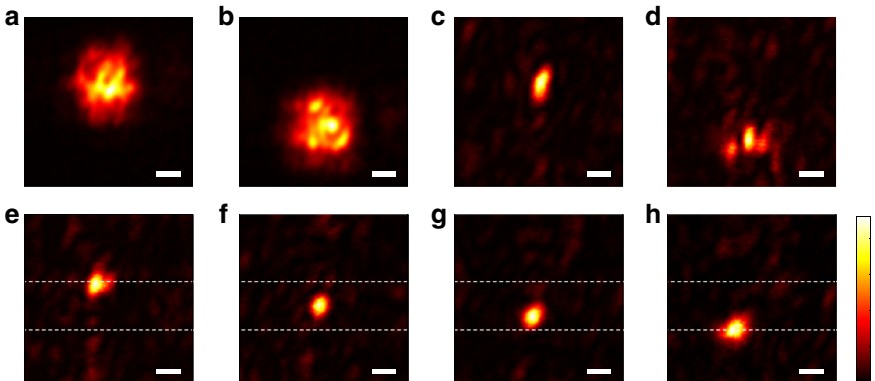

**Fig. 4** Experimental focal scan using two acousto-optic transmissiom matrices: Measured intensity distributions at the ultrasound focal plane for: **a, b** acoustically-tagged light by each of the two ultrasound foci, averaged over 50 random input fields. **c, d** Focusing via SVD of $T_1$ (**c**) or $T_2$ (**d**) independently, as in Figs. 1–2, with corresponding peak to background ratio (PBR) of 102 (**c**) and 83 (**d**). **e–h** Injecting the first singular vector of the matrix $A^\alpha$ (Equation 3), with $\alpha$ values of 0.05 (**e**), 0.1 (**f**), 1 (**g**), and 10 (**h**), demonstrating the ability to scan a tight optical focal spot between the two ultrasound foci. The corresponding PBRs are 80, 97, 100, and 67 for **e, f, g, h** correspondingly. scale-bars, 90 μm

TROVE, the largest variance mode is found by diagonalizing a spatial covariance matrix in the form of $C^H C$, where $C$ is a matrix of output fields measured for random input fields. Such diagonalization is mathematically equivalent to performing an SVD of the matrix $C$. The largest variance mode of TROVE is thus identical to the highest singular valued vector of the AOTM. The only mathematical difference between TROVE and the AOTM is that the matrices used in AOTM are measured using known orthogonal input modes, while in TROVE the input modes are random and unknown. As a result, the two approaches

provide comparable focusing performance. While mathematically similar, the experimental implementation of the AOTM and TROVE is quite different: a practical advantage of the AOTM is that it is not based on a DOPC system, and thus it allows any arbitrary positioning of the SLM and the camera, and any arbitrary pixel-count in each, removing the strict alignment constraints of the DOPC systems required in TROVE and TRUE[35].

In the experimental results of Fig. 4, the FWHM transverse dimensions of the ultrasound foci (Fig. 4a, b) and the sharpest focus obtained by SVD of $A_\alpha$ (Fig. 2f) are 180 μm ± 10 μm, and 50 μm ± 5 μm, respectively. The vertical dimensions of these foci (along the ultrasound axial dimension) are 200 μm ± 20 μm, and 60 μm ± 5 μm, respectively, thus providing here a resolution increase of > ×3.3. The difference between the obtained foci dimensions are attributed to the stability of the experimental system and the limited number of input modes measured and controlled (see Supplementary Notes 5–6).

## Discussion

The AOTM provides a generalized mathematical framework and a useful approach for non-invasive high-resolution optical investigation and light control in complex media. As such, the state-of-the-art acousto-optic guide-star techniques can be readily revisited using the AOTM, including TRUE, iTRUE, and TROVE. While all current acousto-optic based approaches have so far been limited to use the first singular vector of the AOTM, we have shown that lower singular vectors can be exploited to concentrate light around and outside the ultrasound focus. Such minimization of the light intensity inside the acoustic focus may prove useful in e.g., laser therapy applications, when a specific small volume needs to be protected from laser irradiation. This result can also be linked to the recent use of wavefront-shaping for glare reduction[47], but with the AOTM, in a non-invasive fashion. The decomposition of the large ultrasound focus to singular vectors can be linked to the results of all-acoustical DORT experiments on extended reflective targets[48,49], where it has been shown that the singular vectors in free-space correspond to the prolate spheroidal wave-functions[49].

In our proof-of-concept experiments, the speckle grain size was chosen to be considerably larger than the optical wavelength, to allow demonstrating single-speckle scale focusing with the experimentally measured number of input modes $N_{SLM}$, signal to noise ratio and system stability. While the optical speckle grain size constitutes an ultimate limit to the attainable focus size, the actual size of the focus obtained by SVD of the AOTM (and by iTRUE and TROVE) is expected to be larger than the speckle grain when a too small number of input or output modes is measured (i.e., if the measured AOTM is of too small size). The required matrix size for different experimental parameters of speckle grain size and SNR is studied analytically and numerically in Supplementary Notes 5–6 (see Supplementary Figures 6–8). Our analysis suggests that the minimum number of probed modes (both input and output) that is required for single-speckle scale focusing using a single AOTM grows cubically with the ratio between the ultrasound focus diameter and speckle grain size. Thus, the number of required measurements will be considerably larger in experiments in thick diffusive samples where the speckles are of diffraction-limited dimensions. Our theoretical and numerical analyses suggest that for the case of diffraction-limited speckles, even when high frequency ultrasound is used, the number of probed input modes that are required to achieve speckle scale focusing with a single AOTM is of the order of $N_{SLM} \approx 10^6$. With our experimental setup acquisition speed (see below) and a sequential measurements scheme, such a large

number of measurements would be impractically large, and beyond the sample decorrelation time. Given these practical limitations of our setup, and similar to previous works[29], we have chosen to use large speckle grain size, to prove the ability of single-speckle grain focusing.

Concerning the practical timescales for speckle decorrelation in tissue, iterative phase-conjugation (iTRUE) seems more suitable than TROVE or AOTM for single point focusing, as it requires only a few tens of iterations to reach the resolution increase obtained by thousands of TROVE or AOTM measurements. While the total number of measurements (and thus the acquisition time) for focusing at a single position is smaller for iTRUE, it requires repeated iterations for focusing at any different position, and does not allow focus scanning like AOTM or TROVE (see Supplementary Table 1).

The AOTM approach, in its current embodiment, thus presents a novel framework for analysing the performance limits of acousto-optic based light control experiments, but does not allow imaging or focusing in thick scattering biological samples. This fundamental limitation posed by the small speckle grain size, while shared with TROVE and iterative optimization approaches, was not analyzed in previous works, and is importantly highlighted here. Extending acousto-optic approaches to allow optical diffraction-limited imaging in thick scattering samples is thus still a challnge. Potential solutions could involve the joint decomposition of several AOTMs, which effectively form a virtual ultrasound focus that is sharper than the ultrasound beam itself (Figs. 3–4), multiplexed measurements[50], and faster SLMs (see below). The significantly smaller speckle grain dimensions expected in experiments in thick tissue can be alleviated by using longer optical excitation wavelengths[51], and a higher frequency (> 50 MHz) and smaller F-number ultrasound transducer (e.g., a ~ 20 μm diameter 90 MHz ultrasound focus is expected to contain less than 1000 diffraction-limited speckle grains at an illumination wavelength of 1700 nm).

Beyond its conceptual significance, an important practical advantage of the AOTM compared to TRUE, iTRUE, and TROVE, is that the positions of the SLM and camera are arbitrary and no DOPC system nor any careful alignment is required. This also allows an asymmetry in the pixel numbers: measuring on a much larger number of output camera pixels than SLM pixels, as permitted by state-of-the-art cameras. However, a practical disadvantage of the presented acquisition of the AOTM compared to iTRUE is in the slow sequential measurement of input modes. We have used orthogonal phase-only input modes as the basis for our AOTM measurements, however, as with the TM, any set of spanning modes, including amplitude-only inputs can be used to retrieve the AOTM[52].

The experimental system in our experiments was based on a continuous-wave (cw) laser, a liquid crystal SLM with a refresh rate of ~10 Hz, and an ultrasound transducer with 15 MHz central frequency. This choice of available equipment is not optimal for pulsed AOTM measurements in thick tissue, as the laser peak-power and average power are three orders of magnitude lower compared to pulsed lasers[28–30], and the liquid crystal SLM is three to four orders of magnitude slower compared to ferroelectric or MEMS based devices[53,54] or galvanometric scanners[50,55], used for rapid measurement of the TM. These technical limitations restricted our experimental demonstrations to thin static diffusers, which allowed long decorrelation times and sufficiently large scattered light intensity. Since our experiments did not take advantage of any other specific attributes of the thin scatterers, extension to thick multiply-scattering samples may be made possible by using a pulsed laser source and a MEMS based SLMs. However, even with these advancements, given the number of modes required for probing when diffraction-limited speckles

are considered (Supplementary Note 5), the use of this approach is expected to be technically very challenging, and limited to samples with very long decorrelation times, such as ex-vivo or non blood-perfused samples.

The AOTM could be a valuable tool for studying various models of optical propagation in diffusive media, such as the intensity distribution of transmission eigenchannels[40], Anderson localized modes, or the memory-effect inside samples[56,57]. Since the AOTM defines the information content of any acousto-optic experiment, new analysis approaches may be put forward to retrieve more information from within the scattering medium, or generate new focusing modes, e.g., using the generalized Wigner–Smith operator to maximize momentum transfer to a target[58].

The approach can be scaled up to multiple acoustic beams, allowing a large field of view. However, when large volumes are considered, the computational resources for storing the corresponding AOTMs are expected to be very high, and the acquisition procedure very long. This is because the dimension of the AOTM with $L$ ultrasound foci would result in a matrix with $L \times M \times N_{\mathrm{SLM}}$ elements. The size of the matrix, and the acquisition time, may be lowered by replacing the focused ultrasound beams with ultrasonic plane waves, or other parallelized measurements, which can be digitally coherently combined to synthetize various ultrasonic foci in 3D[59,60]. The AOTM could also be extended to non-monochromatic optical illumination, providing spectral, and temporal information on the medium's response, This could be realized by using ultrashort optical pulses[16], or spectrally tunable sources[61], but would results in even higher dimensionality of the AOTM.

## Methods

**Simulations**. For the simulation of an AOTM measurement, two random matrices with Gaussian i.i.d. random amplitude entries and random i.i.d. phase distribution from zero to $2\pi$ were generated to simulate the all-optical transmission matrices between: (1) the SLM plane and the ultrasound focal plane, $T_{\mathrm{SLM\text{-}US}}$, and (2) the ultrasound focal plane and the camera plane, $T_{\mathrm{US\text{-}CAM}}$. The propagation from the SLM to the camera was simulated by multiplying $T_{\mathrm{SLM\text{-}US}}$ by the input-field $E_{\mathrm{in}}$. The ultrasound tagging was simulated by multiplying the field $T_{\mathrm{US\text{-}CAM}}E_{\mathrm{in}}$ pixel-by-pixel by a Gaussian-shaped ultrasound pressure amplitude $P_{\mathrm{US}}$. The result was propagated to the camera by multiplying with $T_{\mathrm{US\text{-}CAM}}$. The number of simulated input and output modes were 4096 and 2048, respectively (Fig. 1b). The simulated random i.i.d. transmission matrices provide a representative model for the diffusive light propagation regime, deep beyond the scattering and transport mean free paths in a multiply-scattering medium, without any correlations (memory effect, ballistic component, etc.). This i.i.d. matrices model is justified since the number of simulated (and experimentally measured) degrees of freedom in such a medium with the small ultrasound tagging volume would be considerably smaller than the total number of optical modes at such depths. This will yield cropped ("filtered") matrices, which are essentially i.i.d.[6], and will not allow observing transport related phenomena such as the bimodal distribution of transmission eigenchannels[62].

**Experiments**. The full experimental setup is described in detail in Supplementary Note 1: a long-coherence continuous-wave (cw) laser at a center wavelength of 810 nm was used as the light source. The laser is a single longitudinal mode, tunable, extended cavity, semiconductor laser, producing up to 1.5 Watt. The maximum power used in our experiments was 300 mW. The laser was provided by DTU Fotonik, Denmark (see acknowledgements).

To measure the ultrasonically modulated light, the laser beam was split to two arms of an interferometer. At the first arm, the laser beam illuminates an SLM (Holoeye Pluto), which is imaged on the first diffuser (Light shaping diffuser, Newport). A spherically focused ultrasonic transducer (V319-SU-F0.75-IN-PTF, Olympus; 15 MHz central frequency, 0.75" focal length, 0.5" element diameter), emits 133 ns long pulses at a center frequency of 15 MHz inside a water filled glass tank. At the second, reference, arm, two acousto-optic modulators (MT-80, AA-Optoelectronics) are used to frequency shift and time gate the reference arm signal, to produce a 133 ns pulse with a central frequency that is shifted by 15 MHz + (5KHz/4) from the original laser frequency. The pulsed reference beam is focused on a mirror placed next to the output plane of the medium (the second diffuser) and is combined with the first beam on a fast camera (Photron Fastcam SA4) operating at $f_{\mathrm{cam}} = 5000$ frames per second, with a 1 microsecond exposure time. To measure the weak ultrasonically tagged field in the presence of the strong untagged background, a double-heterodyne holographic technique was employed[36], combining off-axis and phase-shifting interferometry. To maximize the measurements' signal to noise, the SLM was imaged on the first diffuser surface,

and a Hadamard input basis was used to measure the AOTM[6]. To compensate for any slow phase drifts of the reference arm, a flat-phase mask was displayed on the SLM before each input mode was injected, and the phase of the output field measured with a flat-phase input was subtracted from the phase of each measured output field. The total acquisition time for a single AOTM with 3072 input modes (6144 displayed phase patterns) was limited by the SLM refresh rate (~ 6 Hz), and was 18 min. An iris placed in front of the first diffuser was used to control the speckle size at the target plane. Triggering of all instruments is detailed in Supplementary Figure 2. To minimize the fraction of the unshaped light from the SLM limited fill-factor, a blazed grating phase pattern was displayed on the SLM and the zero diffraction order was blocked. The displayed phase patterns on the SLM were composed of macro pixels of 20×20 physical pixels, displaying the phase sum of the blazed grating and the desired input phase. For direct inspection of the optical fields at the ultrasound focal plane, the second diffuser was removed, and an imaging lens was placed between the camera and the scattering medium. The diffusers used in the double-slice experimental demonstrations were light shaping diffusers (Newport) providing no measurable ballistic components. In the experiments of Fig. 2 the diffuser 1 was composed of a stack of two diffusers, one with a scattering angle of 0.5° and the other 1°. The second slice (between acoustic focus and camera) was a 0.5° light shaping diffuser. In the experiments presented in Fig. 4, both slices were 0.5° light shaping diffusers. The acoustic focus was located at a distance of 10 cm from each of the scattering layers.

**Data analysis**. The experimentally measured AOTMs had 3072 input modes. For each input mode a camera image of the output field with a resolution of 512 × 352 pixels (Fig. 2) or 320 × 240 (Fig. 4) was acquired. To minimize the computational memory requirements, considering the speckle grain size on the camera, one out of three camera pixel in each dimension was taken as an output mode (×3 under-sampling). For the results of Figs. 1–2, SVD of $T$ was performed in Matlab (Mathworks). For the results of Figs. 3–4 SVD was performed on the matrix $A_{\alpha} = \left(T_{1-\alpha 2}^{H} T_{1-\alpha 2}\right)^{+} \left(T_{1+\alpha 2}^{H} T_{1+\alpha 2}\right)$, where $^{+}$ is a Tikhonov regularized pseudo inverse[46]. Since the number of columns and rows of $T$ can be substantially different, the SVD of $B_{\alpha} = \left(T_{1-\alpha 2} T_{1-\alpha 2}^{H}\right)^{+} \left(T_{1+\alpha 2} T_{1+\alpha 2}^{H}\right)$ can be used as well to calculate the optimal focusing input vectors, reducing the size of the analyzed matrix[29].

## Data availability
All relevant data are available from the authors upon request.

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

## Acknowledgements

The authors thank Mingjun Chu and Paul Michael Petersen, Technical University of Denmark, for providing the long-coherence laser source, and Benjamin Judkewitz and Sebastien Popoff for helpful discussions. This project has received funding from the European Research Council (ERC) under the European Union's Horizon 2020 research and innovation program (grants no. 278025, 677909), and LABEX WIFI (Laboratory of Excellence within the French Program "Investments for the Future") under references ANR-10-LABX-24. O.K. was supported by a Marie Curie intra-European fellowship (IEF) and an Azrieli Faculty Fellowship. S.G. acknowledge support from the Institut Universitaire de France (IUF).

## Author contributions

M.F. conceived the idea. O.K. developed the idea, performed modeling and numerical simulations, built the setup, collected data, and performed data analysis. O.K., S.G., F.R., and M.F. contributed to the design of the experiments and to the writing of the manuscript.

## Additional information

**Competing interests:** The authors declare no competing interests.

