## [Peer Review File · Nature Communications]

Reviewers' comments:

Reviewer #1 (Remarks to the Author):

The manuscript "Controlling light in complex media beyond the acoustic diffraction-limit using the acousto-optic transmission matrix" reports the measurement of an optical transmission matrix that specifically describes the propagation of the acoustically modulated optical waves through a scattering medium (named "AOTM") and proposes a relevant framework upon which previous acousto-optic-assisted wavefront shaping approaches can be understood. It also demonstrates the 3-5 fold improvement of focusing resolution compared to the size of acoustic focus by using the first singular vector of AOTM or time-reversal operator (TRO) constructed from two AOTMs. Experimental investigation on optical transmission matrix has been initiated since the work by SM Popoff et. al. (SM Popoff et. al. Physical Review Letters 2010) and has been extensively exploited in the field of wavefront shaping 'through' a scattering medium (SM Popoff et. al. New Journal of Physics 2011, Y Choi et. al. Physical Review Letters 2011, SM Popoff et. al. Physical Review Letters 2011, DB Conkey et.al. Physical Review Letters 2013). Along with the efforts exploiting the transmission matrix, numerous approaches (X Xu et. al. Nature Photonics 2011, Y Wang et. al. Nature Communications 2012, K Si et. al. Nature Photonics 2012, B Judekewitz et. al. Nature Photonics 2013, H Ruan et. al. Scientific Reports 2014) have also been developed for optical focusing and imaging 'inside' a scattering medium based on the optical phase conjugation of acoustically modulated optical waves. This work can be understood as the application of the framework of transmission matrix to the acoustically modulated optical waves in the context of optical focusing inside a scattering medium.

I find the performed experiments are convincing, and the manuscript is clearly written to carefully explain the relevance of this work with the previous studies on optical transmission matrix and acousto-optic-assisted wavefront shaping approaches. However, I have three major concerns regarding the novelty and the significance of this work.

First, it is not surprising that the propagation of acoustically modulated optical waves can be described in a form of matrix equation (as in Eq. (1)) as the acoustic-optic effect here comprises the linear interaction that an acoustic wave coherently disturbs the propagation of optical waves. The acoustically modulated optical waves (for different inputs) have actually been written in the matrix form and similarly exploited in the previous work that is heavily cited in this article (see Methods and Supplementary Information in B Judkewitz et.al. Nature photonics 2013). Also, I find the major principle presented in this work - the interpretation of the results of singular value decomposition (SVD) of AOTM or TRO - is not essentially new. The point that SVD of transmission matrix results in the identification of eigenchannels with high and low energy transmission has been widely known and used in many previous studies (SM Popoff et. al. Physical Review Letters 2010, SM Popoff et. al. Physical Review Letters 2011, M Kim et. al. Nature Photonics 2012). It is also mentioned in a previous study (J Bosch et. al. Optics Express 2016) that the iterative phase conjugation results in the coupling to the open eigenchannel. Furthermore, it is hard to see the difference between the proposed TRO in this work and the variance-encoding matrix (Eq. (5) in Supplementary Information) presented in the previous work of B Judkewitz et.al. Nature photonics 2013.

Second, the experimental geometry is far from a realistic case because the distance between two scattering layers is so large that a single speckle grain size (30 μm) is more than 30 times larger than the laser wavelength (810 nm). Authors claim that they used the geometry for the convenient proof of principle purpose and that the concept of using SVD to create a shaper focus is 'ideally' applicable to any sample geometry where a single speckle grain is much smaller. As authors pointed out, in the ideal condition without any experimental noise, SVD would identify the high and low transmission eigenchannels that are associated with propagating modes confined at the center and edge of Gaussian-shaped acoustic focus. However, in a realistic situation, the relative size of a speckle grain and an acoustic focus is much important parameter. When a single speckle grain becomes much smaller than an acoustic focus, neighboring speckle grains 'feel' (i.e. will be modulated by) similar acoustic pressure because the Gaussian-shaped acoustic pressure field is much slowly varying compared to the optical field. In this situation, experimental noise (i.e.

shot noise, sample stability, distortion in acoustic field, and so on) easily exceeds the signal resulting from slight differences in acoustic pressure, leading to the first eigenchannel being composed of multiple speckle grains. The effect of noise is already observable from the experimental results in this article (V3 and V4 in Figure 2 have a significant leakage to the speckle grain at the center of acoustic focus, thereby not presenting the exact ring pattern predicted in the simulation). So, I strongly suggest the authors provide either the experimental results of wavelength-scale focusing resolution or in-depth numerical studies on the effect of noise to support that the method is generally applicable for any experimental geometry with smaller speckle grains.

Lastly, there is more or less no improvements in focusing or imaging performance compared to the previous relevant works on SVD and TRO – two main operators presented in this work. This is because the experimental technique and geometry, and the way to handle the measured complex field is fundamentally the same with the previous works. The resolution improvement factor of 3-5 is also more or less the same with the resolution improvement factor reported in H Ruan et. al. Scientific Reports 2014 (equivalent to SVD in this paper) and B Judekewitz et. al. Nature Photonics 2013 (equivalent to TRO in this paper). Furthermore, while the previous studies demonstrate the practical significance of improved focusing resolution with an additional imaging experiments, this work is lack of such practical significance. Also, the advantage of AOTM in that the camera and SLM can be arbitrary is marginal compared to the advantage of a large number of measurable/controllable optical modes in an optical phase conjugation system.

In general, I find the work is lack of novelty and practical significance compared to the previous studies of conventional transmission matrix and acousto-optic-assisted wavefront shaping techniques. Especially, SVD and TRO has been proposed in previous studies (e.g. H Ruan et. al. Scientific Reports 2014 and B Judekewitz et. al. Nature Photonics 2013) with almost identical experimental setup and geometry. I agree that the proposed framework of AOTM serves here as a new interpretation of already proposed operators, but this wouldn't be significant for the readership of Nature Communications.

This work may be reconsidered when there is some major update on experimental results or principle such as showing experimental results of wavelength-scale focusing resolution (when a single speckle grain is much smaller than acoustic focus) or proposing unique capability of AOTM, rather than interpreting the wavefront shaping methods that are previously proposed, in the context of deep tissue imaging or focusing.

Reviewer #2 (Remarks to the Author):

In this work, a generalized model of acousto-optic wavefront tagging based on an acousto-optic transmission matrix is presented. This framework connects the optical wavefront defined at the plane of a spatial light modulator (SLM) located outside of a scattering medium, to the frequency-shifted optical wavefront located at a camera also located outside of a scattering medium, via a linear operator. By doing so, the presented framework indirectly models the light that must have traveled through an ultrasound focus located within a scattering medium to become frequency-shifted. This clever model is experimentally tested and verified to show focusing beyond the acoustic diffraction limit.

The model presented here is new and useful and the experimental results appear sound. Based on the included details, I think a skilled researcher could reproduce this work. In my view, the model and results do not significantly improve upon prior demonstrations based on TRUE, iTRUE and TROVE. But the framework does offer minor improvements – for example, mitigating the need to precisely align the SLM and camera. Furthermore, the model is new and interesting to the optics community in its own right and will likely lead to additional developments in the future. Based on these reasons, I would argue for publication.

Here are some more detailed comments that should be addressed:

- The authors should try to make it more clear up front, near Eq. 2, that the subscript m denotes spatial locations at the plane of a camera located outside of the scattering medium (i.e., m is not connected at all to the spatial coordinates in the ultrasound focus). This only became clear to me after carefully looking at Fig. 1.
- Page 4 paragraph 2: "(PBR) of the formed focus" – this is the focus in the medium, not at the camera pixel, right?
- Page 4 paragraph 3: This all assumes a Gaussian-type structure for the acoustic focus? What if the acoustic focus is non-Gaussian?
- Is there a specific reason why 15 MHz was used here? Previous demonstrations use higher frequencies, and it would seem that a smaller acoustic focus would require less measurements and thus offer a faster experiment.
- Comparison with TROVE: it is of note here that this technique displays a specific wavefront basis on the SLM for each new measurement, whereas TROVE randomly scrambles the incident wavefront for each new measurement. Here no SLM-camera alignment is needed, whereas alignment is needed for TROVE. It would be nice to see some sort of discussion/analysis of this.

Reviewer #3 (Remarks to the Author):

The manuscript discusses the use of a matrix formalism that describes the propagation of light interacting with an acoustic focus in the linear regime. The combination of light and ultrasound is especially promising in the context of imaging in turbid tissue. The AOTM that the authors introduce is a useful formalism to think about concepts in linear acousto-optics, and is likely to be useful in the case of media that are not strongly dynamic. While that is a limitation, it is not a very severe one, as in explorative research there are enough model systems that are not strongly time dependent.

The manuscript is well written and informative. I have a few concerns that could be addressed in a revision.

First, the necessary dimensionality of the matrix is related to the geometry (size of the focus, wavelength) and is nicely calculated in the appendix. Formula S14 however may be incomplete as one would expect the rms noise to enter quadratically as e.g. in DOI: 10.1364/BOE.4.001759. More importantly the conclusions of the appendix, namely that the minimum number of matrix rows/columns needed for effective imaging scales as the cube of the US/optical wavelength ratio, are discussed explicitly in the main text.

This scaling limits the use of the full TM for e.g. biological imaging somewhat, as the full measurement plus processing time even for a single image point involves measuring thousands of frames, during which the sample has to be stable well within an optical wavelength. I think the author's comments in the discussion that their system is not optimized for biological imaging is somewhat evasive, and it is not entirely obvious without any calculation that "extension to thick multiply scattering tissue can be made possible...".

In conclusion, the manuscript and research overall are of very high quality and I recommend publication after the authors have addressed the above comments.

Reviewers' comments:

Reviewer #1 (Remarks to the Author):

The manuscript “Controlling light in complex media beyond the acoustic diffraction-limit using the acousto-optic transmission matrix” reports the measurement of an optical transmission matrix that specifically describes the propagation of the acoustically modulated optical waves through a scattering medium (named “AOTM”) and proposes a relevant framework upon which previous acousto-optic-assisted wavefront shaping approaches can be understood. It also demonstrates the 3-5 fold improvement of focusing resolution compared to the size of acoustic focus by using the first singular vector of AOTM or time-reversal operator (TRO) constructed from two AOTMs.

Experimental investigation on optical transmission matrix has been initiated since the work by SM Popoff et. al. (SM Popoff et. al. Physical Review Letters 2010) and has been extensively exploited in the field of wavefront shaping ‘through’ a scattering medium (SM Popoff et. al. New Journal of Physics 2011, Y Choi et. al. Physical Review Letters 2011, SM Popoff et. al. Physical Review Letters 2011, DB Conkey et.al. Physical Review Letters 2013). Along with the efforts exploiting the transmission matrix, numerous approaches (X Xu et. al. Nature Photonics 2011, Y Wang et. al. Nature Communications 2012, K Si et. al. Nature Photonics 2012, B Judkewitz et. al. Nature Photonics 2013, H Ruan et. al. Scientific Reports 2014) have also been developed for optical focusing and imaging ‘inside’ a scattering medium based on the optical phase conjugation of acoustically modulated optical waves. This work can be understood as the application of the framework of transmission matrix to the acoustically modulated optical waves in the context of optical focusing inside a scattering medium.

I find the performed experiments are convincing, and the manuscript is clearly written to carefully explain the relevance of this work with the previous studies on optical transmission matrix and acousto-optic-assisted wavefront shaping approaches.

However, I have three major concerns regarding the novelty and the significance of this work.

First, it is not surprising that the propagation of acoustically modulated optical waves can be described in a form of matrix equation (as in Eq. (1)) as the acoustic-optic effect here comprises the linear interaction that an acoustic wave coherently disturbs the propagation of optical waves.

We agree that since, fundamentally, any linear system can be represented in a matricial form, one may state that the matrix formalism is “not surprising” for any system. We believe that it is also true for the original optical transmission-matrix (TM) work [Popoff et al. PRL 2010, cited over 700 times], and its many illuminating outcomes, published in a large set of follow-up works. Some examples for such follow-up works include the photoacoustic-TM [Chaigne et al. Nature Photonics, 2013], time-gated TM [Keng et al. Nature Photonics 2015, Badon et al. Nature Photonics 2016, Mounaix et al. Phys. Rev. A (2017)], and even a work published just earlier this year on the use of SVD of the time-gated TM [Seungwon Jeong et al. Nature Photonics, 2018]. We believe that, in this field of matrix approaches, our work is original and gives rise to important contributions as :

- 1) The formulation of the acousto-optical transmission matrix as a general description of any acousto-optical experiment.
- 2) Demonstration of the experimental measurement and application of the AOTM in an setup that is not based on a digital optical phase-conjugation (DOPC) system.
- 3) The formulation, analysis, and comparison of all acousto-optic experiments in a single framework, including, in the revised manuscript, a theoretical analysis of the resolution limitations of these techniques. In particular, for the first time, this theoretical and numerical performance limitations analysis includes the regime of diffraction-limited speckle grains (not considered, for example, in the TROVE work)
- 4) The first analysis and experimental realization of lower-valued singular vectors of the AOTM, with important potential implications for controlled minimization of laser intensity in controlled volumes.
- 5) The connection between focusing using the first singular vector of the AOTM and the results of iTRUE and TROVE, and the proof for their fundamental equivalent nature.

The acoustically modulated optical waves (for different inputs) have actually been written in the matrix form and similarly exploited in the previous work that is heavily cited in this article (see Methods and Supplementary Information in B Judkewitz et.al. Nature photonics 2013).

While the TM of the scattering medium, as presented by Popoff et al. was used in the analysis of the TROVE work, the matrices measured in TROVE were not the AOTM, but were only matrices of output speckle patterns measured for unknown random inputs. While similar in dimensionality, one should not confuse these matrices with the AOTM: unlike any transmission-matrix, the matrices measured in the TROVE work do not represent the Green's functions of the medium but rather only its output for unknown random inputs. It is important to emphasize that the measurements performed in the TROVE work cannot allow the direct retrieval of the AOTM, which is characteristic of the medium.

Since this point may be indeed confusing for the informed reader, we added a clarification on this delicate point to the revised manuscript. This statement reads:

“It is also important to distinguish between the output speckles matrix, C , measured in TROVE and the AOTM: unlike the AOTM, the TROVE matrix C does not represent the Green's functions of the medium but rather only its output for unknown random inputs. Thus, the measurements performed in TROVE do not allow the direct retrieval of the AOTM, which is characteristic of the medium.”

In addition, prior to our work, since the AOTM formalism was not put forward, any light control beyond the acoustic diffraction limit in acousto-optics was thought to require the use of a DOPC system. Our work allow to go beyond this conceptual barrier, but also removes the practical requirement of constructing a DOPC system for such experiments. Using the matrix formalism to consider the works of iTRUE, TROVE and others, and connecting them under the same framework provides a new way to achieve these results, to surpass them (e.g. by utilizing low singular values

vectors), or to understand their fundamental limitations (as we do in the new supplementary section presented below). We believe that the AOTM formalism will allow for even more new possibilities based on the above.

Also, I find the major principle presented in this work - the interpretation of the results of singular value decomposition (SVD) of AOTM or TRO – is not essentially new. The point that SVD of transmission matrix results in the identification of eigenchannels with high and low energy transmission has been widely known and used in many previous studies (SM Popoff et. al. Physical Review Letters 2010, SM Popoff et.al. Physical Review Letters 2011, M Kim et. al. Nature Photonics 2012). It is also mentioned in a previous study (J Bosch et. al. Optics Express 2016) that the iterative phase conjugation results in the coupling to the open eigenchannel.

We respectfully disagree with the above claim: unlike the ‘conventional’ TM, the SVD of the AOTM does NOT give access to the transmission eigenchannels of the system. It may be indeed confusing since the mathematical decomposition procedure is the same, but the physical interpretation of the SVD of the AOTM is fundamentally different than the SVD of the normal TM. For instance, the first singular values (SV) of a system maximizes the energy transmission between the sources or modulators and the receptors. In the case of the TM the sources are in the plane of the SLM, so before the medium, and the camera is after, so the SVD gives the transmission eigenchannels. In contrast, for the AOTM, the camera records the light at the modulated frequency, so the source for the signal are located in the US focus. The first SV gives the best generation of tagged photon, but not necessarily the best transmission through the whole system. Thus, the SVs of the AOTM are not transmission eigen-channels of the scattering system. In particular, it depends on the shape of the ultrasound focus, as is apparent in the comparison between the distribution of SVs of the conventional TM and the AOTM that is given in Figure 1c of the original manuscript. The purpose of this figure, presented also below, was exactly to avoid this confusion between the SVD of the TM and that of the AOTM, and specifically display the differences in their physical interpretations.

(Fig. 1c from the original manuscript, showing the first 1,000 singular values of the AOTM (blue), which are dictated by the ultrasound focus shape, vs. the first 1,000 singular values of an all-optical TM measured without ultrasound tagging (gray), which follows the general Marčenko Pastur distribution)

Additionally, while it is true that the SVD of the *optical TM* was used in many previous works to identify transmission eigenchannels, as we acknowledge in our original manuscript (“SVD is a powerful tool in matrix analysis, and was recently used to identify transmission-eigenchannels³⁹⁻⁴², and for selective focusing^{19, 27”}), and the connection between iterative time-reversal and SVD was put forward more than two decades before the work of Bosch et al. in the works of Prada and Fink in Ultrasound [e.g. Prada et al. Wave-motion, 1994], the use of SVD of any transmission-matrix for diffraction-limited focusing on extended targets was never performed before. In the context of scattering media, injecting the first singular vectors of the TM (i.e. the transmission eigenchannels) does NOT correspond to focusing at all. This is evident, for example, in the very recent work by Seungwon Jeong et al. published earlier this year in *Nature Photonics* [Seungwon Jeong et al. *Nature Photonics*, 2018], where injection of singular vectors into a scattering samples resulted in enhanced energy transmission to extended targets but not to focusing below the target size, as was also the case in the work of Popoff et al. [PRL2011], and previous works in ultrasound. The novel fact that the SVD of the AOTM can allow super-resolved focusing, without the presence of discrete point-like targets, is achieved only because the unique profile of the ultrasound focus is not flat, but Gaussian-shaped (see comparisons to a top-hat target in Supplementary Figures S4-S5). This is also a novel result in the context of the inspiring works in ultrasound, considering extended targets, but not reporting on sharp focusing below the target size [Robert and Fink, JASA, 2009].

To better clarify this point, and avoid the confusion between the SVD of the AOTM and the SVD of the ‘conventional’ all-optical TM, we have added the two following explanations to the revised manuscript, in addition to a reference to the results from the recent work of Jeong et al. (reference [22] in the revised manuscript):

“While the singular value decomposition (SVD) of an all-optical TM was used to focus light with the DORT technique on reflective targets¹⁸⁻²¹, such all-optical or all-acoustical approaches are not capable of focusing light to the diffraction limit in multiple scattering samples unless an isolated, strongly-reflective, and point-like target is present^{16, 19}. Moreover, even in the presence of such a reflective target, the obtained focus is of the target dimensions, and not the desired optical diffraction-limit^{22”}

“The fact that the SVD of the AOTM provides sharp focusing is unique, and is due to the gaussian-like shape of the ultrasound tagging focus. This is in contrast to the SVD of the all-optical TM that yields transmission-eigenchannels, whose injection into a scattering medium does not lead to sharp focusing^{22”}.

Furthermore, it is hard to see the difference between the proposed TRO in this work and the variance-encoding matrix (Eq. (5) in Supplementary Information) presented in the previous work of B Judkewitz et.al. Nature photonics 2013.

Our AOTM measurement approach and setup is technically and conceptually very different than the variance-encoding (TROVE) measurement approach. The differences are both conceptual, where in TROVE one is trying to maximize the

variance of the ultrasound-encoded light, and in the AOTM we are measuring a set of Green's functions of the sample in the presence of ultrasound tagging, and also practical: in the substantially different experimental setup and measurement procedure. Another important difference is that while in TROVE one is only maximizing the variance, thus using only the first singular vectors of the measured matrices, in our work we were not limited to such an analysis. In particular, we have shown that not only the first singular value of the AOTM is interesting to observe, but also the lower singular values. These lead to formation of ring-like focusing structures that concentrate the light *outside* and around rather than inside the ultrasound focus. The theoretical and experimental observation and realization of such unique modes inside complex samples has not been suggested or observed before, and is one of the novel contribution of our work. As we note in our discussion, these may be interesting for protecting a small selected volume from optical irradiation in biomedical contexts. In essence, the AOTM contains all the information about the propagation of light through the acoustically excited area. To draw an analogy, this difference between analyzing only the first singular vector (as in TROVE) to the consideration of other singular vectors is similar to the difference between the first iterative based focusing experiments by Vellekoop and Mosk and the works based on the measurement and analysis of the TM.

The technique we present thus introduces a framework to tailor or measure the properties of the modulated optical field, not being limited to focusing or even imaging applications.

That said, we agree with the referee that while our approach was motivated by the DORT works in acoustics, and developed on its principles of the use of SVD for avoiding the necessity of iterative phase-conjugation, a careful comparison of the end result has indeed shown that the SVD of the AOTM focusing approach is mathematically very similar (though not equivalent) to the TROVE approach. We believe that our discovery of this fundamental connection between two a-priori very different approaches, which are based on different assumptions, is enlightening, sheds new light on possible focusing mechanisms, and may inspire following works focused on either sub-wavelength focusing, imaging or transmission-eigenchannels.

To better explain these differences in initial assumptions and goals, we have added the following explanation to the revised manuscript:

“While interestingly the final focusing results are similar, the principles leading to focusing via AOTM and TROVE are different: in the AOTM (and iTRUE, and DORT) a mode that maximizes the transmission through the ultrasound focus is sought after, while in TROVE, a mode that maximizes the variance of the encoded light is resolved. While different in initial goals, as shown above, these two modes are, in fact, equivalent, shedding light on the focusing mechanism.”

Second, the experimental geometry is far from a realistic case because the distance between two scattering layers is so large that a single speckle grain size (30 μm) is more than 30 times larger than the laser wavelength (810 nm). Authors claim that they used the geometry for the convenient proof of principle purpose and that the

concept of using SVD to create a sharper focus is 'ideally' applicable to any sample geometry where a single speckle grain is much smaller. As authors pointed out, in the ideal condition without any experimental noise, SVD would identify the high and low transmission eigenchannels that are associated with propagating modes confined at the center and edge of Gaussian-shaped acoustic focus. However, in a realistic situation, the relative size of a speckle grain and an acoustic focus is much important parameter. When a single speckle grain becomes much smaller than an acoustic focus, neighboring speckle grains 'feel' (i.e. will be modulated by) similar acoustic pressure because the Gaussian-shaped acoustic pressure field is much slowly varying compared to the optical field. In this situation, experimental noise (i.e. shot noise, sample stability, distortion in acoustic field, and so on) easily exceeds the signal resulting from slight differences in acoustic pressure, leading to the first eigenchannel being composed of multiple speckle grains. The effect of noise is already observable from the experimental results in this article (V3 and V4 in Figure 2 have a significant leakage to the speckle grain at the center of acoustic focus, thereby not presenting the exact ring pattern predicted in the simulation). So, I strongly suggest the authors provide either the experimental results of wavelength-scale focusing resolution or in-depth numerical studies on the effect of noise to support that the method is generally applicable for any experimental geometry with smaller speckle grains.

We thank the referee for pointing out this very important point, which was not properly thoroughly addressed in the original manuscript. We strongly agree regarding the importance of the speckle grain size in relation to the acoustic focus size and SNR for effective focusing in practical thick multiply-scattering samples.

We wish to note that the effect of smaller speckle grain size was indeed already analyzed, both theoretically and numerically, in the original manuscript's supplementary sections 5-6, where we have shown that the theoretically required minimum number of measurements (input modes, N_{SLM}) grows cubically with the ratio between the ultrasound focus diameter and the speckle grain size, *even in the absence of noise*. This important fundamental limitation of acousto-optical based focusing (including TROVE and iterative optimization) has not been addressed and have been largely ignored in previous works. In particular, the results obtained in the TROVE work were only possible since large speckle grains, an order of magnitude larger than the wavelength, were used in this work, not discussing the strict limitation that it poses on any practical use of TROVE in practical imaging scenarios.

In order to not dispose or hide this important limitation, we have included in our original manuscript, the detailed theoretical and numerical analysis of the expected focus size and enhancement for smaller speckle grain size. Particularly, in the original supplementary section 6 we have presented the numerical simulation results that are attached below (Supplementary Figure S7), which show that for smaller speckle grains, the obtained focus size contains multiple speckle grains if the size of the measured matrix remains unchanged, *even without the presence of noise*. Our original simulation results of Supplementary Figure S7 already showed that for a ratio of ultrasound focus to speckle grains size of $\sigma_{\text{us}}/\sigma_{\text{speckle}}=21$, the obtained focus size with a single AOTM is only 3-4 times smaller than the ultrasound focus diameter, i.e. the diameter of the obtained focus is approximately 7 times larger than the speckle grain

size. In our original Supplementary Figure S7(f,g) shown below, we have also explicitly displayed the obtained foci, formed by multiple speckle grains.

Supplementary Figure S7: simulation results for focal spot size obtained with the AOTM in the noise-less case, using 10,000 output modes (camera pixels), compared to TRUE focusing: a. AOTM focus spot diameter (σ_{AOTM}) as a function of N_{SLM} , for different ultrasound focus diameter ($\sigma_{\text{US}}=4.2$ to 21 speckle grains diameter, σ_{speckle}); Vertical scale is normalized to the ultrasound focus diameter size, σ_{US} ; **b.** Same as (a) for TRUE focusing, showing the acoustic diffraction limitation; **(d-i)** Example of the obtained intensity distribution at the ultrasound focal plane for a single realization of the largest simulated ultrasound focus, having a diameter of ~ 21.3 speckle grains, i.e. containing ~ 450 speckle grains (marked by * in (a)), for: **d.** flat SLM; **e.** TRUE focusing; **f-i,** injecting the first four singular vectors of the AOTM measured with $N_{\text{SLM}}=10,000$. The experiments of Figure 2 were obtained with an ultrasound focus size of $\sigma_{\text{US}}\approx 6.5\sigma_{\text{speckle}}$. The simulation results for $\sigma_{\text{US}}\approx 7\sigma_{\text{speckle}}$ are given by the bold black line in (a-c). To achieve single speckle grain focusing in these conditions $N_{\text{SLM}}>\sim 2,500$ is required

As we explain in the original Supplementary Section 5, the reason that even in the noiseless case speckle-grain scale focusing is not achievable if an insufficient number of modes are measured in the AOTM, is due to the statistical noise of the projected and detected speckle patterns.

While we have studied in depth the consideration arising from smaller speckle grain size, we have indeed not analyzed with similar depth the effects of finite measurement noise. In our original Supplementary information, we have only given the expected theoretical effect of measurements noise on the required number of measurements, but have not performed in-depth numerical studies of these effects. Following the referee suggestions we have performed an in-depth numerical study of the effects of measurements noise for both the cases of the experimental parameters, as well as the case of diffraction-limited speckle grains tagged by high-frequency ultrasound. The results of the new study are presented in the *new Supplementary Figure S8* (see below). In addition, we rewrote and extended the *theoretical* derivations of Supplementary Section 5, to include specifically the expected focus size as a function of all system parameters, including speckle grain size and measurement SNR. To minimize the length of this reply, we will not copy here the full new Supplementary Section 5, but only present here one graph (*Supplementary Figure S6*) plotting the theoretical limit on the expected focus size as a function of speckle grain size and number of measured degrees of freedom. This result explicitly show that the focus size achieved by a single AOTM analysis (and TROVE), while smaller than the

ultrasound focus, cannot reach the single speckle grain scale unless a large enough number of input mode measurements are acquired. As stated above, this derivation is general, and encompasses the limits of AOTM, TROVE and also iterative optimization with acousto-optic feedback [e.g. Tay et al. Sci.Reports, 2014)].

Supplementary Figure S6: Theoretical lower bound for the focal spot size obtained with the AOTM first singular vector in the noise-less case, using 10,000 output modes (camera pixels). The plotted focal spot size is estimated by $r_{\text{focus}} = \sqrt{1 + (2\Delta r)^2}$, where Δr is taken from Supplementary Equation S14. The asterisk (*) marks the parameters of our experiments with single AOTM focusing.

A brief summary of the new *numerical* results for various SNRs is given in the new Supplementary Figure S8 (please refer to the revised Supp.Information for full details):

Supplementary Figure S8: Numerically obtained focus size in the case of finite SNR and small speckle grain:
a. AOTM focus spot diameter (σ_{AOTM}) as a function of the ratio between the ultrasound focus and the speckle grain size ($\sigma_{US}/\sigma_{speckle}$), for different noise-to-signal (NSR) ratios. **(b-d)** Comparison of the intensity distribution at the focus for the case of $\sigma_{US}/\sigma_{speckle}=50$ between TRUE focusing (b), and the first (c) and fourth (d) singular vectors of the AOTM, for NSR=0; **(e-g)** same as (b-d) for NSR=0.1; **(h-j)** same as (e-g) for $\sigma_{US}/\sigma_{speckle}=7$.

The final result of the new theoretical analysis for the effects of SNR is given by Supplementary Equation S18 (see revised supplementary information for the new full derivation):

“

$$\Delta r > \sim \sqrt{\frac{3.6}{N_{SLM}(1-NSR^2)} \left(\frac{\sigma_{us}}{\sigma_{speckle}} + \frac{NSR}{\sqrt{2M}} \left(\frac{\sigma_{us}}{\sigma_{speckle}} \right)^2 \right)} \sigma_{us} \quad (S18)$$

where M is the number of measurements used to determine the ultrasonically-tagged energy.

Measurement noise thus affect the focus size by two terms: the first is the lower enhancement, which interestingly, even for SNR as low as SNR=3, is reduced only by a factor of $(1 - NSR^2) \approx 0.9$. The second is the second term in (S18),

where the finite SNR becomes non-negligible when: $\frac{NSR}{\sqrt{2M}} \left(\frac{\sigma_{us}}{\sigma_{speckle}} \right)^2 > \frac{\sigma_{us}}{\sigma_{speckle}}$, i.e.

when: $SNR < \frac{1}{\sqrt{2M}} \frac{\sigma_{us}}{\sigma_{speckle}} = \sqrt{\frac{N_{speckles}}{2M}}$. This suggests a higher SNR is required in the case of smaller optical speckle grain size (compared to the ultrasound focus dimension). “

As is clear from the above analytical derivation and numerical results, achieving speckle-scale focusing with diffraction limited speckles is not possible at reasonable acquisition times with the current experimental embodiment of the approach. Importantly, the exact same limitation occurs for TROVE, which marks the current state-of-the-art in acousto-optic guided focusing. Thus, in both the TROVE work and in our work, speckle grains considerably larger than the diffraction limit were chosen for the experimental demonstration. However, unlike all previous works, we have explicitly discussed and analyzed these fundamental limitations, for the first time to our knowledge. In order to make this point perfectly clear, we have added the following discussion to the revised manuscript.

“Our analysis suggests that the required minimum number of probed modes (input and output) for speckle-scale focusing using a single AOTM grows cubically with the ratio between ultrasound focus diameter and speckle grain size. Thus the number of required measurements will be considerably larger in experiments in thick samples, where the speckles are diffraction-limited. Thus, it is not surprising that the obtained focusing performance in the various approaches are similar. Our theoretical and numerical analysis suggests that for the case of diffraction-limited speckles, even when high frequency ultrasound is used, the number of input modes required to achieve speckle scale focusing with a single AOTM is of the order of $N_{\text{SLM}} \approx 106$. With our experimental setup acquisition rate (see below) and the sequential measurements scheme, such a large number of measurements would be impractically large and beyond the sample decorrelation time. Given these practical limitations of our setup, and similar to previous works³², we have chosen to use large speckle grain size, to prove the concept of single-speckle grain focusing beyond the ultrasound diffraction limit.

Importantly, since the AOTM provides a general description of all acousto-optical experiments, the performance limitations on focus size and the requirements for SNR and matrix dimensions are fundamentally identical to TROVE as well as for other acousto-optic based focusing approaches, such as iterative optimization based focusing³⁶. The AOTM approach, in its current embodiment, thus presents a novel framework for analysis and performance of acousto-optic based light control experiments, but does not allow imaging or focusing in thick scattering biological samples. This fundamental limitation posed by small speckle grain size, while shared with TROVE and iterative optimization approaches was not analyzed before and is highlighted for the first time here. Extending these approaches to allow optical diffraction-limited focusing in thick scattering samples is challenging. Potential solutions should involve the joint decomposition of several AOTMs that effectively form a virtual ultrasound focus that is sharper than the ultrasound focus (Figures 3-4), multiplexed measurements⁵², and faster SLMs (see below). ”

“Since our experiments did not take advantage of any other specific attributes of the thin scatterers, extension to thick multiply scattering samples may be made possible by using a pulsed laser source and a MEMS based SLMs. However, even with these advancements, given the number of modes required for probing when diffraction-limited speckles are considered (Supplementary Section 5), the use of this approach is expected to be technically very challenging, and limited to

samples with very long decorrelation times, such as ex-vivo or non blood-perfused samples.”

We sincerely believe that our novel approach and framework, in addition to the new analysis of limitations of all acousto-optical techniques to date marks an important contribution to the field, and also correct the expectations for speckle-scale focusing using acousto-optic guidance, which were not analyzed in previous works, in particular in TROVE.

Lastly, there is more or less no improvements in focusing or imaging performance compared to the previous relevant works on SVD and TRO – two main operators presented in this work. This is because the experimental technique and geometry, and the way to handle the measured complex field is fundamentally the same with the previous works. The resolution improvement factor of 3-5 is also more or less the same with the resolution improvement factor reported in H Ruan et. al. Scientific Reports 2014 (equivalent to SVD in this paper) and B Judkewitz et. al. Nature Photonics 2013 (equivalent to TRO in this paper). Furthermore, while the previous studies demonstrate the practical significance of improved focusing resolution with an additional imaging experiments, this work is lack of such practical significance.

While we agree with the referee that the obtained focusing performance is similar to the ones obtained by the current state of the art approaches, we respectfully disagree with the statement that the experimental technique and geometry is fundamentally the same as in previous works: as is explained in our manuscript, while all techniques to date relied on time-reversal of acoustically modulated light, our work proves from first principles that a time-reversal/phase-conjugation capability is *not* required to achieve sub-diffraction focusing. We show to obtain optimal focusing using energy criteria, by maximizing the generation of tagged photon using the knowledge of the AOTM. This leads to a totally different measurement geometry, where the SLM and camera are arbitrary located. This is in contrast to the precise alignment of SLM and camera that is mandatory in TRUE/iTRUE/TROVE experiments, for making sure that the computed modes are reinjected into the medium in precisely the same output channels. Our approach allows us to directly compute the phase mask on the SLM that will focus light at the desired position.

Moreover, this allows us to measure, study and utilize new features of the ultrasound tagging, allowed by the low singular valued vectors of the AOTM, which are impossible to reach via simple phase-conjugation.

Perhaps most importantly, our work shows that the obtained focusing performance are not a limitation of the specific chosen focusing technique, but rather a fundamental limitations provided by dimensionality of the measured matrix (number of measured input modes), and the focusing performance of all techniques is inherently limited by this fundamental limit, overlooked by all previous works. The goal of our work was more fundamental than showing a specific focusing technique with improved capabilities compared to current approach, but rather to present a new framework for all acousto-optical experiments that allows to understand the fundamental limitations

of acousto-optic guided focusing and imaging. We believe that this marks an important contribution to the field, and will be used as the tool of choice for developing and analyzing future works involving acousto-optic tagging.

To clarify this point we have added the following statement to the revised manuscript:

“Importantly, since the AOTM provides a general description of all acousto-optical experiments, the performance limitations on focus size and the requirements for SNR and matrix dimensions are fundamentally identical to TROVE as well as for other acousto-optic based focusing approaches, such as iterative optimization based focusing³⁶.”

Also, the advantage of AOTM in that the camera and SLM can be arbitrary is marginal compared to the advantage of a large number of measurable/controllable optical modes in an optical phase conjugation system.

We agree with the referee that for practical applications the number of *measurable* and *controllable* modes is at least as important as is the advantage of decoupling the SLM and camera. However, the referee statement that the number of measurable/controllable modes in phase-conjugation system is larger than in the AOTM approach is not accurate: Since the measurements in AOTM are performed by a high resolution camera the number of *measurable* modes in the AOTM approach is identical to the number of *measurable* modes in TRUE/iTRUE/TROVE. Moreover, while the number measurable modes in DOPC systems is limited by the number of SLM pixels (effectively limited to approximately 10^5 in the state of the art works employing DOPC), the AOTM approach is not limited to such symmetry, and the number of measured modes could be orders of magnitudes larger than the number of SLM pixels, as permitted by 10s of megapixels modern cameras. This important fact, which provides important advantage in e.g. low SNR conditions was mentioned in our original manuscript: “This also allows an asymmetry in the pixel numbers: measuring on a much larger number of output camera pixels than SLM pixels, as permitted by state-of-the-art cameras.”

In addition, while it is true that for focusing to a single point the number of controllable modes in a single-shot measurement of a phase-conjugation system is orders of magnitude larger than with the AOTM, and an iTRUE approach is expected to yield faster sharper focusing, this is not the case when focusing to *multiple* sub-wavelength foci is required, such as is done for focus scanning in potential imaging experiments: In such a scenario the total number of measurements required using a phase-conjugation iTRUE/TROVE system or the AOTM is similar, as we analyze in Supplementary table 1. The fundamental reason for this interesting result is that for the TROVE focusing approach, the important dimension of the variance matrix is dictated by the number of speckle realizations (number of matrix rows), rather than the number of camera/SLM pixels of the DOPC system (number of columns), since the latter is considerably smaller than the former (1,000 vs. >100,000 in the TROVE experiments). In the iTRUE experiments, a large number of total measurements is required, since the iterative measurements have to be repeated for each different focus location, rather than calculated from four acoustic-foci AOTM measurement, which

can also be measured in parallel via e.g. chirped ultrasound modulation or plane wave illumination. This was stated in our original manuscript in relation to the AOTM dimensions. Following the referee comment, we have added to this statement also the acquisition time:

“The size of the matrix, **and the acquisition time**, may be lowered by replacing the focused ultrasound beams with ultrasonic plane waves, **or other parallelized measurements**”

In general, I find the work is lack of novelty and practical significance compared to the previous studies of conventional transmission matrix and acousto-optic-assisted wavefront shaping techniques. Especially, SVD and TRO has been proposed in previous studies (e.g. H Ruan et. al. Scientific Reports 2014 and B Judekewitz et. al. Nature Photonics 2013) with almost identical experimental setup and geometry..

We have addressed this point in our answers above. Shortly: unlike all previous works, based on a DOPC experimental setup, our experiments have a very different geometry and experimental setup, that is not allowed by previous works. We respectfully disagree regarding the lack of novelty as we present not just a novel analysis but also the first prediction, measurements and synthesis of unique ring-like modes, which are the low singular values modes of the AOTM. Beyond these, the main practical significance of the AOTM is in the elimination for the need of a DOPC system, but most importantly, our work is aimed at a more fundamental impact for the analysis of waves in complex media, and for the fundamental limitations on their probing and control.

I agree that the proposed framework of AOTM serves here as a new interpretation of already proposed operators, but this wouldn't be significant for the readership of Nature Communications

As we explain in our manuscript and our answers above, our work presents much more than new interpretations of already proposed operators. The new interpretations is just a starting point and related to the first singular values. All the rest of the work, including lower singular-value analysis, decoupling the SLM and camera, providing fundamental limits on focus size control, etc. is novel.

This work may be reconsidered when there is some major update on experimental results or principle such as showing experimental results of wavelength-scale focusing resolution (when a single speckle grain is much smaller than acoustic focus) or proposing unique capability of AOTM, rather than interpreting the wavefront shaping methods that are previously proposed, in the context of deep tissue imaging or focusing.

Following the constructive and important comments of Referee #1 we have performed new theoretical analysis and in-depth numerical studies for the performance of the technique in the case of diffraction-limited speckle grain size, and under various signal-to-noise conditions. These are given in the revised Supplementary sections 5-6.

These results mark the limits not only of our technique but also to any technique making use of acousto-optic guidance.

As shown in these analyses, performing experiments with diffraction-limited speckle grains would require acquisition time that is longer than the decorrelation times of our samples, with the current embodiment of the experiments. Importantly, the same statement is correct for the experiments of TROVE, and as we show, to any acousto-optic technique. Thanks to the constructive comments of the referee we now state this limitations very clearly in our revised manuscript. We wish to emphasize that this limitation as the reason that focusing to the diffraction-limited speckle grain dimensions has not been reported by *any* work to date, and thus we believe that the request from us to perform such an experiments is not needed for this publication, and currently out of reach of current experiments worldwide.

Regarding the referees request to present a unique capability of the AOTM: the presented use of any input and output modes is unique to the AOTM, as is the lack of requirement for a DOPC system. The new control of low singular values modes is also a novel prediction and result. We believe that these novel features, in addition to the fundamental connections to operators such as the Wigner-Smith operator, proposed in our works would be very interesting to a wide audience, in particular for those readers from a more fundamental background or from related fields that are not particularly only interested in focusing and imaging applications. A point of view which we seem to share with Referees #2 and #3.

Reviewer #2 (Remarks to the Author):

In this work, a generalized model of acousto-optic wavefront tagging based on an acousto-optic transmission matrix is presented. This framework connects the optical wavefront defined at the plane of a spatial light modulator (SLM) located outside of a scattering medium, to the frequency-shifted optical wavefront located at a camera also located outside of a scattering medium, via a linear operator. By doing so, the presented framework indirectly models the light that must have traveled through an ultrasound focus located within a scattering medium to become frequency-shifted. This clever model is experimentally tested and verified to show focusing beyond the acoustic diffraction limit.

The model presented here is new and useful and the experimental results appear sound. Based on the included details, I think a skilled researcher could reproduce this work. In my view, the model and results do not significantly improve upon prior demonstrations based on TRUE, iTRUE and TROVE. But the framework does offer minor improvements – for example, mitigating the need to precisely align the SLM and camera. Furthermore, the model is new and interesting to the optics community in its own right and will likely lead to additional developments in the future. Based on these reasons, I would argue for publication.

We thank the referee for his positive view of our work. As we explain in our reply to Referee #1, we share Referee #2 exact view of the significance of our work as a new framework and model to all acousto-optically guided focusing, allowing new information retrieval and studies, and not as a specific technique for improved focusing.

Here are some more detailed comments that should be addressed:

- The authors should try to make it more clear up front, near Eq. 2, that the subscript m denotes spatial locations at the plane of a camera located outside of the scattering medium (i.e., m is not connected at all to the spatial coordinates in the ultrasound focus). This only became clear to me after carefully looking at Fig. 1.

To clarify this point we have added the following clarification in the paragraph following Eq.2

“Importantly, r_m , is located at the camera plane outside of the scattering medium. “

- Page 4 paragraph 2: “(PBR) of the formed focus” – this is the focus in the medium, not at the camera pixel, right?

This is correct. In order to clarify this point we have rephrased the statement regarding the PBR in the manuscript text to:

“The peak to background ratio (PBR) of the formed focus at the ultrasound focus plane inside the scattering medium is also...”

- Page 4 paragraph 3: This all assumes a Gaussian-type structure for the acoustic focus? What if the acoustic focus is non-Gaussian?

We thank the referee for this very interesting question. Indeed, the nature of the singular modes is dependent on the shape of the ultrasonic focus. In our original manuscript we provide one numerical example for a different shaped focus in Supplementary Figures S3 and S5. Referred to in the main text as:

“For the Gaussian-shaped ultrasound focus considered in Fig.1 the singular values decrease gradually. Different shaped ultrasound foci would result in different distributions (see a numerical example in Supplementary section 4). “

Following the referees question and a question raised by referee #1, we have also added the following explanation to this paragraph and the revised manuscript:

“This eigenvector is, by definition, the first singular vector of the AOTM, T_u , providing the sharpest optical focus. The fact that the SVD of the AOTM provides sharp focusing is unique, and is due to the gaussian-like shape of the ultrasound tagging focus. This is in contrast to the SVD of the all-optical TM that yields transmission-eigenchannels, whose injection into a scattering medium does not lead to sharp focusing²²”

- Is there a specific reason why 15 MHz was used here? Previous demonstrations use higher frequencies, and it would seem that a smaller acoustic focus would require less measurements and thus offer a faster experiment.

This is perfectly correct: Higher ultrasound frequency transducers would allow us to use smaller speckle grains and to potentially shorten our acquisition time. Unfortunately, such transducers and related signal generation equipment were not available for our experiments. One may argue that 15Mhz frequency provides a tradeoff between low frequency (few MHz) probes, which provide the greatest focusing/imaging depth but with very large acoustic focus area, and high frequency ones (>40 MHz), which indeed provide smaller focusing, but limit the working depth a few millimeters. However, the main reason for the use of 15MHz frequency in our experiments was the availability of the related equipment, and our desire to perform proof-of-principle experiments of our novel approach, which were conveniently scaled versions of those that can be performed at high frequencies. In order to discuss the possible advantage of high frequency ultrasound, we included the following discussion to the revised manuscript

The significantly smaller speckle grain dimensions expected in experiments in thick tissue can be alleviated by using longer optical excitation wavelengths⁵³, and a higher frequency (50-100MHz) and smaller F-number ultrasound transducer (e.g. an ~20 μm diameter 90MHz ultrasound focus is expected to contain less than 1,000 diffraction-limited speckle grains at an illumination wavelength of 1700nm), which were not experimentally available in our experiments.

- *Comparison with TROVE: it is of note here that this technique displays a specific wavefront basis on the SLM for each new measurement, whereas TROVE randomly scrambles the incident wavefront for each new measurement. Here no SLM-camera alignment is needed, whereas alignment is needed for TROVE. It would be nice to see some sort of discussion/analysis of this.*

The reviewer is indeed correct. Previous techniques are based on time reversing the acoustically modulated field, and necessitated a DOPC system with a wavefront shaping setup that has to be precisely aligned with the holographic measurement setup, in order to send back the phase conjugated light in the exact same channels. Our focusing technique relies on finding the wavefront that maximizes the generation of tagged photons, which can be extracted directly from the AOTM. As long as the light modulated by the SLM interacts with the acoustic focus area (the general case of a scattering medium), the AOTM allows us to focus light. Thus, no precise alignment of the SLM is required. In particular, the SLM and the camera do not need to be aligned with respect to one another. While this was stated several times in our original manuscript, in order to deepen the discussion on this comparison we have added the following discussion to the revised manuscript:

“While interestingly the final focusing results are similar, the principles leading to focusing via AOTM and TROVE are different: in the AOTM (and iTRUE, and DORT) a mode that maximizes the transmission through the ultrasound focus is sought after, while in TROVE, a mode that maximizes the variance of the encoded light is resolved. While different in initial goals, as shown above, these two modes are, in fact, equivalent, shedding light on the focusing mechanism. It is also important to distinguish between the output speckles matrix, C , measured in TROVE and the AOTM: unlike the AOTM, the TROVE matrix C does not represent the Green functions of the medium but rather only its output for unknown random inputs. Thus, the measurements performed in TROVE do not allow the direct retrieval of the AOTM, which is characteristic of the medium. Beyond this important fundamental difference, a practical advantage of the AOTM is that any arbitrary positioning of the SLM and camera can be used to produce the sub acoustic-diffraction optical focus, removing the alignment constraints of a DOPC system of TROVE and TRUE”

To discuss the interesting point raised by the referee for the potential use of random input modes rather than orthogonal basis vectors for probing the medium we have added the following statement to the revised manuscript:

“We have used orthogonal input modes as the basis for our AOTM measurements, however, as with the TM, any set of spanning modes, including random, and amplitude-only inputs can be used to retrieve the AOTM⁵⁴.”

Where we have also added a reference for such a recent work:

“54. Drémeau, A. et al. Reference-less measurement of the transmission matrix of a highly scattering material using a DMD and phase retrieval techniques. *Optics express* 23, 11898-11911 (2015).”

Reviewer #3 (Remarks to the Author):

The manuscript discusses the use of a matrix formalism that describes the propagation of light interacting with an acoustic focus in the linear regime. The combination of light and ultrasound is especially promising in the context of imaging in turbid tissue. The AOTM that the authors introduce is a useful formalism to think about concepts in linear acousto-optics, and is likely to be useful in the case of media that are not strongly dynamic. While that is a limitation, it is not a very severe one, as in explorative research there are enough model systems that are not strongly time dependent.

We thank the referee for his general positive view of our work.

The manuscript is well written and informative. I have a few concerns that could be addressed in a revision.

First, the necessary dimensionality of the matrix is related to the geometry (size of the focus, wavelength) and is nicely calculated in the appendix. Formula S14 however may be incomplete as one would expect the rms noise to enter quadratically as e.g. in DOI: 10.1364/BOE.4.001759.

We thank the referee for pointing us to the suggested reference, which we were unaware of. Indeed, our original analysis did not take into account the reduced enhancement expected for low SNR conditions. To address this point we have rewritten this supplement section, and included the correct expression for the expected enhancement in the presence of noise, adding a reference to the above work. This revised section reads:

“

5.3 Analysis of obtained focus size for the finite SNR case:

The above derivation assumed noise-free measurements. Measurement noise may be considered by changing the intensity enhancement to $\eta = \frac{\pi}{4} N_{SLM} (1 - NSR^2)$, where $NSR = (SNR)^{-1}$ is the noise-to-signal ratio⁹, and adding an additional noise term $n \approx \frac{NSR}{\sqrt{M}} \cdot E_{speckle} (\sigma_{us}/\sigma_{speckle})^2$ to the right-hand side of equation (S12), where M is the number of measurements used to determine the ultrasonically-tagged energy:

$$\eta E_{speckle} \frac{\Delta r^2}{2\sigma_{us}^2} > (B_2 - B_1) + n \quad (S16)$$

$$\eta E_{speckle} \frac{\Delta r^2}{2\sigma_{us}^2} > \sqrt{2} E_{speckle} \left(\frac{\sigma_{us}}{\sigma_{speckle}} \right) + \frac{NSR}{\sqrt{M}} \cdot E_{speckle} \left(\frac{\sigma_{us}}{\sigma_{speckle}} \right)^2 \quad (S17)$$

$$\Delta r > \sim \sqrt{\frac{3.6}{N_{SLM}(1-NSR^2)} \left(\frac{\sigma_{us}}{\sigma_{speckle}} + \frac{NSR}{\sqrt{2M}} \left(\frac{\sigma_{us}}{\sigma_{speckle}} \right)^2 \right)} \sigma_{us} \quad (S18)$$

Measurement noise thus affect the focus size by two terms: the first is the lower enhancement, which interestingly, even for SNR as low as SNR=3, is reduced only by a factor of $(1 - NSR^2) \approx 0.9$. The second is the second term in (S18), where the finite

SNR becomes non-negligible when: $\frac{NSR}{\sqrt{2M}} \left(\frac{\sigma_{us}}{\sigma_{speckle}} \right)^2 > \frac{\sigma_{us}}{\sigma_{speckle}}$, i.e. when: $SNR <$

$\frac{1}{\sqrt{2M}} \frac{\sigma_{us}}{\sigma_{speckle}} = \sqrt{\frac{N_{speckles}}{2M}}$. This suggests a higher SNR is required in the case of smaller optical speckle grain size (compared to the ultrasound focus dimension).

It is important to note that the above analysis for the obtained focus dimensions with and without noise considers the case of focusing using a single AOTM. Using two or more AOTMs for focusing is expected to provide smaller focus size, and is advantageous over focusing using a single AOTM, since the decomposition of several AOTMs provides an effective virtual acoustic focus that is sharper than each of the individual, acoustic-diffraction limited foci, as is visible in the experimental results of Figure 4(c-h).

“

Where reference [9] is the above work by Yilmaz et al.

More importantly the conclusions of the appendix, namely that the minimum number of matrix rows/columns needed for effective imaging scales as the cube of the US/optical wavelength ratio, are (NOT?) discussed explicitly in the main text. This scaling limits the use of the full TM for e.g. biological imaging somewhat, as the full measurement plus processing time even for a single image point involves measuring thousands of frames, during which the sample has to be stable well within an optical wavelength. I think the author's comments in the discussion that their system is not optimized for biological imaging is somewhat evasive, and it is not entirely obvious without any calculation that "extension to thick multiply scattering tissue can be made possible..."

We completely agree with the referee, and apologize if this point has not been emphasized sufficiently in the original manuscript. To make it perfectly clear that the AOTM approach, while general, is not appropriate for dynamic samples, in particular in biomedical contexts, compared to phase-conjugation approaches, we have added the following clarifications to the supplementary section of the revised manuscript:

“Our analysis suggests that the required minimum number of probed modes (input and output) for speckle-scale focusing using a single AOTM grows cubically with the ratio between ultrasound focus diameter and speckle grain size. Thus the number of required measurements will be considerably larger in experiments in thick samples, where the speckles are diffraction-limited. Thus, it is not surprising that the obtained focusing performance in the various approaches are similar. Our theoretical and numerical analysis suggests that for the case of diffraction-limited speckles, even when high frequency ultrasound is used, the number of input modes required to achieve speckle scale focusing with

a single AOTM is of the order of $N_{\text{SLM}} \approx 106$. With our experimental setup acquisition rate (see below) and the sequential measurements scheme, such a large number of measurements would be impractically large and beyond the sample decorrelation time. Given these practical limitations of our setup, and similar to previous works³², we have chosen to use large speckle grain size, to prove the concept of single-speckle grain focusing beyond the ultrasound diffraction limit.

Importantly, since the AOTM provides a general description of all acousto-optical experiments, the performance limitations on focus size and the requirements for SNR and matrix dimensions are fundamentally identical to TROVE as well as for other acousto-optic based focusing approaches, such as iterative optimization based focusing³⁶. The AOTM approach, in its current embodiment, thus presents a novel framework for analysis and performance of acousto-optic based light control experiments, but does not allow imaging or focusing in thick scattering biological samples. This fundamental limitation posed by small speckle grain size, while shared with TROVE and iterative optimization approaches, was not analyzed before and is highlighted for the first time here. Extending these approaches to allow optical diffraction-limited focusing in thick scattering samples is thus challenging. Potential solutions should involve the joint decomposition of several AOTMs, which effectively form a virtual ultrasound focus that is sharper than the ultrasound focus (Figures 3-4), multiplexed measurements⁵², and faster SLMs (see below). ”

“Since our experiments did not take advantage of any other specific attributes of the thin scatterers, extension to thick multiply scattering samples may be made possible by using a pulsed laser source and a MEMS based SLMs. However, even with these advancements, given the number of modes required for probing when diffraction-limited speckles are considered (Supplementary Section 5), the use of this approach is expected to be technically very challenging, and limited to samples with very long decorrelation times, such as ex-vivo or non blood-perfused samples.”

In conclusion, the manuscript and research overall are of very high quality and I recommend publication after the authors have addressed the above comments.

We thank the referee for his positive view of our work and its significance.

Reviewers' comments:

Reviewer #1 (Remarks to the Author):

In review of the revised manuscript and response letter, I found that the authors have addressed most of my comments and meaningfully improved the quality of the manuscript. As described in the authors' response, the new aspects of this work in comparison with the previous works are the demonstration of singular vectors over the full range of singular values and the discovery of ringshaped pattern of lower-valued singular vectors. Also, the rigorous theoretical analysis on the resolution limitation of various acousto-optic experiments is an important new addition. This provides an insight that has not been thoroughly studied in the previous studies. Therefore, I am inclined to recommend the publication if these new aspects are clearly highlighted and the concern described below is addressed in the main text of the manuscript. I am still concerned with the way authors compare their work with the previous iterative phase conjugation and TROVE works in the manuscript. Authors stated that AOTM approached is technically and conceptually different from the previous works. I have difficulty in accepting this this statement. AOTM has very similar property to the matrix measured in the TROVE work in the regards that both matrices are essentially a collection of output complex field for different input basis. The only difference is whether the input basis is known or not - in AOTM, different input bases are synthesized using SLM, and in the matrix of TROVE work, the basis are randomly realized using a rotating diffuser. The important question here would be "what is the unique benefit of AOTM (or knowing the input basis) in controlling light beyond acoustic diffraction limit?", compared to the previous works based on phase conjugation. Authors presented two operators for controlling light beyond acoustic diffraction limit – (1) SVD of single AOTM and (2) SVD of the generalized TRO using two AOTMs (please note here that it would be clearer to specify the proposed generalized TRO is for two ultrasound beams in the session title, instead of multiple). Essentially, authors organized the manuscript in a way that the results from (1) and (2) comprise the main results: The results supporting (1) is shown in Fig. 1 and Fig.2 and the results associated with (2) is shown in Fig. 3 and Fig. 4. (1) As authors described, the first singular vector of a single AOTM, which has the sharpest spatial profile among other singular vectors, provides the same result with the iterative phase conjugation reported in [K Si et. al. Scientific Reports 2012 and H Ruan et. al. Scientific Reports 2014]. The difference lies in the way to implement it. In fact, in the practical aspect, iterative phase conjugation is much efficient method as it only requires a few tens of measurement and writing to converge to the wavefront solution. (2) As I originally addressed in my previous comment, the Eq. (5) in Supplementary Information of TROVE work of [B Judkewitz et.al. Nature photonics 2013] is very similar to the SVD of the generalized TRO using two AOTMs in Eq. (3) of this paper. Authors response, "Our AOTM measurement approach and setup is technically and conceptually very different than the variance-encoding (TROVE) measurement approach. ..." misses this point. This statement explains the difference between SVD of single AOTM and SVD of generalized TRO using two AOTMs, and not the difference between SVD of generalized TRO using two AOTMs and TROVE. Authors added the following paragraph to the revised manuscript to describe the distinction between AOTM and TROVE. "While interestingly the final focusing results are similar, the principles leading to focusing via AOTM and TROVE are different: in the AOTM (and iTRUE, and DORT) a mode that maximizes the transmission through the ultrasound focus is sought after, while in TROVE, a mode that maximizes the variance of the encoded light is resolved. While different in initial goals, as shown above, these two modes are, in fact, equivalent, shedding light on the focusing mechanism." This statement is vague and does not clearly

convey the relation of their proposed operators and the previous works in acousto-optics. My suggestion is that authors consider my arguments given above and revise the manuscript in such a way to fairly compare the present work with the previous works in terms of conceptual novelty and experimental results. This will ensure the delivery of the correct message to the scientific community on the unique contribution of this work.

Reviewer #2 (Remarks to the Author):

I'd like to thank the authors for their detailed comments and responses. All of my concerns were addressed, and I think the authors also did an adequate job addressing the concerns of the other reviews as well in this rewritten version. I would argue for publication.

We have revised our manuscript according to the referees' suggestions. We wish to thank the referees for their thorough and general positive reviews.

In our reply below we cite the referees' comments in *italics*, our replies appear in standard text, and the corresponding revisions made to the manuscript appear in blue. The revisions made to the manuscript are highlighted in yellow in the manuscript text.

Sincerely,

Mathias Fink

Reviewers' comments:

Reviewer #1:

In review of the revised manuscript and response letter, I found that the authors have addressed most of my comments and meaningfully improved the quality of the manuscript. As described in the authors' response, the new aspects of this work in comparison with the previous works are the demonstration of singular vectors over the full range of singular values and the discovery of ringshaped pattern of lower-valued singular vectors. Also, the rigorous theoretical analysis on the resolution limitation of various acousto-optic experiments is an important new addition. This provides an insight that has not been thoroughly studied in the previous studies. Therefore, I am inclined to recommend the publication if these new aspects are clearly highlighted and the concern described below is addressed in the main text of the manuscript.

We thank the referee for his positive view of the novel aspects of our work. In order to better highlight these important novel contributions of our work we have revised the abstract of our manuscript, which now reads:

Here, we introduce the acousto-optic transmission matrix (AOTM), which is an ultrasonically-encoded, spatially-resolved, optical scattering-matrix. The AOTM provides both a generalized framework to describe any acousto-optic based technique, and a tool for light control and focusing beyond the acoustic diffraction-limit inside complex samples. We experimentally demonstrate complex light control using the AOTM singular vectors, and utilize the AOTM framework to analyze the resolution limitation of acousto-optic guided focusing approaches.

We also added the following sentence to the introduction section:

We also utilize the AOTM framework to analyze the resolution limitation of SVD-based acousto-optic guided focusing approaches.

We wish to note that in addition to the use of lower singular vectors and the theoretical analysis for the resolution-limits of acousto-optic based approaches, the AOTM approach does not require a DOPC system, which is the basis for the previous works of both iTRUE and TROVE.

I am still concerned with the way authors compare their work with the previous iterative phase conjugation and TROVE works in the manuscript. Authors stated that AOTM approached is technically and conceptually different from the previous works. I have difficulty in accepting this this statement. AOTM has very similar property to the matrix measured in the TROVE work in the regards that both matrices are essentially a collection of output complex field for different input basis. The only difference is whether the input basis is known or not - in AOTM, different input bases are synthesized using SLM, and in the matrix of TROVE work, the basis are randomly realized using a rotating diffuser.

We thank the referee for reiterating this important point. Indeed, while the initial considerations for developing AOTM and TROVE are different, and technically their experimental implementations are distinct (AOTM not requiring a DOPC system or any phase-conjugation system, thus enabling arbitrary positioning of the camera and SLM), the mathematical SVD operations employed in both techniques are equivalent, except for the random vs. orthonormal basis. We also agree that strictly in terms of focusing, ignoring the use of low singular-vectors, the results obtained by the AOTM can be identically obtained via TROVE or iTRUE. We are thus happy to revise the manuscript to make these points clearer. In order to do so we have rewrote the relevant paragraph to read:

Interestingly, the analogy between the results obtained by SVD of the AOTM and those obtained with TROVE is not coincidental. The underlying reason is that the variance maximization approach of TROVE is, in fact, based on performing an SVD to a measured matrix of acoustically-modulated output modes that is very similar to the AOTM: in TROVE, the largest variance mode is found by diagonalizing a spatial covariance matrix in the form of $C^H C$, where C is a matrix of output fields measured for random input fields. Such diagonalization is mathematically equivalent to performing an SVD of the matrix C . The largest variance mode of TROVE is thus identical to the highest singular valued vector of the AOTM. The only mathematical difference between TROVE and the AOTM is that the matrices used in AOTM are measured using known orthogonal input modes, while in TROVE the input modes are random and unknown. As a result, the two approaches provide comparable focusing performance. While mathematically similar, the experimental implementation of the AOTM and TROVE is quite different: a practical advantage of the AOTM is that it is not based on a DOPC system, and thus it allows any arbitrary positioning of the SLM and the camera, and any arbitrary pixel-count in each, removing the strict alignment constraints of the DOPC systems required in TROVE and TRUE³⁵

The important question here would be “what is the unique benefit of AOTM (or knowing the input basis) in controlling light beyond acoustic diffraction limit?”, compared to the previous works based on phase conjugation.

We agree that a very interesting question is what is the unique benefit of using random vs. using known *and orthogonal* input modes. Practically, since the number of speckles in each input mode is large, and the number of controlled modes is only a small fraction of the total number of participating modes, the random input modes of TROVE can be considered to be practically orthogonal, and thus the main difference between the two approaches is the requirement of DOPC system in TROVE, as mentioned in our reply, and in the revised paragraph, above.

Authors presented two operators for controlling light beyond acoustic diffraction limit – (1) SVD of single AOTM and (2) SVD of the generalized TRO using two AOTMs (please note here that it would be clearer to specify the proposed generalized TRO is for two ultrasound beams in the section title, instead of multiple).

Following the referee’s request, we have changed the title of the relevant section to ‘**two ultrasound beams**’ instead of ‘multiple ultrasound beams’

*Essentially, the authors organized the manuscript in a way that the results from (1) and (2) comprise the main results: The results supporting (1) is shown in Fig. 1 and Fig.2 and the results associated with (2) is shown in Fig. 3 and Fig. 4.
(1) As authors described, the first singular vector of a single AOTM, which has the sharpest spatial profile among other singular vectors, provides the same result with the iterative phase conjugation reported in [K Si et. al. Scientific Reports 2012 and H Ruan et. al. Scientific Reports 2014]. The difference lies in the way to implement it. In fact, in the practical aspect, iterative phase conjugation is much efficient method as it only requires a few tens of measurements and writing to converge to the wavefront solution.*

We thank the referee for raising this point. We agree that considering practical speckle decorrelation timescales, and for single-point focusing, iterative phase conjugation is

advantageous over AOTM or TROVE. If multiple foci positions are required and speckle decorrelation is not a limiting factor, TROVE and AOTM may require lower number of total measurements than iTRUE, as is analyzed in our Supplementary Table 1. However, this is indeed not an expected scenario in most biomedical imaging applications.

In order to highlight this important practical aspect we have added the following statement to the Discussion section:

Concerning the practical timescales for speckle decorrelation in tissue, iterative phase-conjugation (iTRUE) seems more suitable than TROVE or AOTM for single point focusing, as it requires only a few tens of iterations to reach the resolution increase obtained by thousands of TROVE or AOTM measurements.

(2) As I originally addressed in my previous comment, the Eq. (5) in Supplementary Information of TROVE work of [B Judkewitz et.al. Nature photonics 2013] is very similar to the SVD of the generalized TRO using two AOTMs in Eq. (3) of this paper. Authors response, "Our AOTM measurement approach and setup is technically and conceptually very different than the variance-encoding (TROVE) measurement approach. ..." misses this point. This statement explains the difference between SVD of single AOTM and SVD of generalized TRO using two AOTMs, and not the difference between SVD of generalized TRO using two AOTMs and TROVE. Authors added the following paragraph to the revised manuscript to describe the distinction between AOTM and TROVE. "While interestingly the final focusing results are similar, the principles leading to focusing via AOTM and TROVE are different: in the AOTM (and iTRUE, and DORT) a mode that maximizes the transmission through the ultrasound focus is sought after, while in TROVE, a mode that maximizes the variance of the encoded light is resolved. While different in initial goals, as shown above, these two modes are, in fact, equivalent, shedding light on the focusing mechanism." This statement is vague and does not clearly convey the relation of their proposed operators and the previous works in acousto-optics. My suggestion is that authors consider my arguments given above and revise the manuscript in such a way to fairly compare the present work with the previous works in terms of conceptual novelty and experimental results. This will ensure the delivery of the correct message to the scientific community on the unique contribution of this work.

To clarify this point, we have performed the following modifications to the revised manuscript:

First, we rewrote the paragraph introducing the generalized TRO using two AOTMS, which now reads:

AOTM using two ultrasound beams

The first singular vector of a single AOTM (Figs.1-2) allows focusing only at a single point located at the center of the ultrasonic focus. Given the large number of measurements required for such focusing, this forms a limitation for the use of this approach for applications such as imaging. However, as was recently shown by Judkewitz et al. in their TROVE work²⁹, a joint analysis of several matrices of acousto-optically modulated output modes, measured for different ultrasound foci positions, can allow focus scanning. Here, we apply the same mathematical approach to jointly decompose multiple AOTMs measured with different ultrasound focus position. Specifically, we exploit the joint analysis of two AOTMs to perform a scan of a tight optical focus over multiple positions.

We wish to note that our original manuscript already clearly stated, at several instances, that the presented approach is directly adapted from TROVE, two examples include:

“following the multi-focus analysis of TROVE²⁹, we used the first eigenvectors or singular vectors of the following generalized TRO, to form a focus at different controlled positions“
and

“This result can be extended to allow scanning in two or three dimensions using more ultrasound beams, as was demonstrated by Judkewitz et al. using four ultrasound foci²⁹.”

Secondly, to fully address the referee’s concern regarding the clarity of our statements comparing AOTM and TROVE, cited in his text above, we have removed this statement and rewrote this paragraph, which now reads:

Interestingly, the analogy between the results obtained by SVD of the AOTM and those obtained with TROVE is not coincidental. The underlying reason is that the variance maximization approach of TROVE is, in fact, based on performing an SVD to a measured matrix of acoustically-modulated output modes that is very similar to the AOTM: in TROVE, the largest variance mode is found by diagonalizing a spatial covariance matrix in the form of $C^H C$, where C is a matrix of output fields measured for random input fields. Such diagonalization is mathematically equivalent to performing an SVD of the matrix C . The largest variance mode of TROVE is thus identical to the highest singular valued vector of the AOTM. The only mathematical difference between TROVE and the AOTM is that the matrices used in AOTM are measured using known orthogonal input modes, while in TROVE the input modes are random and unknown. As a result, the two approaches provide comparable focusing performance. While mathematically similar, the experimental implementation of the AOTM and TROVE is quite different: a practical advantage of the AOTM is that it is not based on a DOPC system, and thus it allows any arbitrary positioning of the SLM and the camera, and any arbitrary pixel-count in each, removing the strict alignment constraints of the DOPC systems required in TROVE and TRUE³⁵

Reviewer #2:

I'd like to thank the authors for their detailed comments and responses. All of my concerns were addressed, and I think the authors also did an adequate job addressing the concerns of the other reviews as well in this rewritten version. I would argue for publication.

We thank the referee for his positive view of our work and its significance.